**RESEARCH**                                                                    **Open Access**

# Transcription shapes DNA replication initiation to preserve genome integrity

Yang Liu[1,2†], Chen Ai[1†], Tingting Gan[1,2†], Jinchun Wu[1,2], Yongpeng Jiang[1,2], Xuhao Liu[1], Rusen Lu[1,2], Ning Gao[2,3], Qing Li[2,4], Xiong Ji[1,2] and Jiazhi Hu[1,2*]

* Correspondence: hujz@pku.edu.cn
†Yang Liu, Chen Ai and Tingting Gan contributed equally to this work.
[1]The MOE Key Laboratory of Cell Proliferation and Differentiation, School of Life Sciences, Genome Editing Research Center, Peking University, Beijing 100871, China
[2]Peking-Tsinghua Center for Life Sciences, Peking University, Beijing 100871, China
Full list of author information is available at the end of the article

## Abstract

**Background:** Early DNA replication occurs within actively transcribed chromatin compartments in mammalian cells, raising the immediate question of how early DNA replication coordinates with transcription to avoid collisions and DNA damage.

**Results:** We develop a high-throughput nucleoside analog incorporation sequencing assay and identify thousands of early replication initiation zones in both mouse and human cells. The identified early replication initiation zones fall in open chromatin compartments and are mutually exclusive with transcription elongation. Of note, early replication initiation zones are mainly located in non-transcribed regions adjacent to transcribed regions. Mechanistically, we find that RNA polymerase II actively redistributes the chromatin-bound mini-chromosome maintenance complex (MCM), but not the origin recognition complex (ORC), to actively restrict early DNA replication initiation outside of transcribed regions. In support of this finding, we detect apparent MCM accumulation and DNA replication initiation in transcribed regions due to anchoring of nuclease-dead Cas9 at transcribed genes, which stalls RNA polymerase II. Finally, we find that the orchestration of early DNA replication initiation by transcription efficiently prevents gross DNA damage.

**Conclusion:** RNA polymerase II redistributes MCM complexes, but not the ORC, to prevent early DNA replication from initiating within transcribed regions. This RNA polymerase II-driven MCM redistribution spatially separates transcription and early DNA replication events and avoids the transcription-replication initiation collision, thereby providing a critical regulatory mechanism to preserve genome stability.

**Keywords:** DNA replication initiation, Transcription, MCM redistribution, Transcription-replication initiation collision, Genome instability, DNA damage

## Background

The mammalian genome is categorized into active and inactive compartments linked to key cellular processes according to chromatin activity and histone modifications, and transcription mainly occurs in active chromatin compartments [1–4]. Early DNA replication is initiated within active chromatin compartments, followed by elongation into inactive compartments [5]. Therefore, early DNA replication and gene

transcription both occur within active chromatin compartments, raising the pivotal question of how these processes are spatially and temporally coordinated to avoid replication-transcription collisions and subsequent DNA damage [6–9]. Previous studies suggest that transcription might affect DNA replication initiation, but the mechanism underlying this interaction in mammalian cells remains elusive [10–15].

DNA replication occurs throughout the entire genome, in contrast to transcription, and tracing early DNA replication requires precise identification of genomic DNA replication origins and initiation zones. Replication origins have been well characterized in bacteria and yeast, but their locations in mammalian cells remain unclear [16, 17]. Many attempts have been made to identify replication origins and initiation zones in mammalian cells. Chromatin immunoprecipitation-sequencing (ChIP-seq) of pre-replication complex (pre-RC) components is difficult because high-quality antibodies against the origin recognition complex (ORC) and mini-chromosome maintenance complex (MCM) are not available [18]. Repli-seq has been applied in studies exploring the determination of replication timing, which have suggested that replication domains range from 400 to 800 kb in size [5, 19]. Recently, high-resolution Repli-seq was employed to profile initiation zones by dissecting the S phase into 16 fractions [20]. The dNTP synthesis inhibitor hydroxyurea (HU) can induce replication stress to slow replication elongation and thus greatly improve the resolution of EdU-marked DNA replication initiation (EdU-seq-HU) experiments [13, 15]. Small nascent DNA strands (SNS) have been used to map DNA replication origins [21–23]. Moreover, Okazaki fragments (OK) generated during DNA replication have also been employed to identify initiation zones that harbor one or more origin(s) within each zone [10, 24]. These methods reveal that early DNA replication initiation occurs in active chromatin compartments in which transcription also occurs [7].

The process of replication initiation requires origin recognition by the ORC, MCM loading, and MCM activation [6, 25]. MCM complexes are loaded as inactive double hexamers in a process dependent on the ORC during the late M and G1 phases. When a cell enters the S phase, a proportion of the MCM complexes are activated to begin unwinding double-stranded (ds)DNA, initiating two replication forks that move bidirectionally [26–28]. The ORC stably binds chromatin, while ring-shaped MCM double hexamers encircling dsDNA may slide along the chromatin, uncoupling the ORC and MCM [29–32]. MCM redistribution has been reported in *Drosophila melanogaster*, *Xenopus* egg extract, and asynchronized human HeLa cells and 2fTGH cells, but the mechanism underlying this process remains unexplored [33–35]. RNA polymerases are among the most likely candidates responsible for MCM redistribution. T7 RNA polymerase can force yeast MCM complexes to fall off linear dsDNA in vitro [36]. In addition, RNA polymerase II is also able to cause MCM relocation at ribosomal DNA loci in budding yeast [37]. However, MCM redistribution by the transcription machinery in mammalian cells has not been confirmed experimentally [32, 35].

Here, we developed a new assay, nucleoside analog incorporation loci sequencing (NAIL-seq), to precisely map genome-wide early replication initiation zones (ERIZs). ERIZs are primarily located in non-transcribed regions in open chromatin compartments, which are mutually exclusive with transcription elongation regions. Furthermore, inhibition of transcription leads to MCM redistribution and early replication, but not ORC re-localization, in transcribed regions. Failure by cells to prevent DNA

replication initiation in transcribed regions leads to gross DNA damage. Therefore, we propose that transcription regulates early DNA replication initiation by redistributing MCM complexes to non-transcribed regions to avoid replication-transcription collisions and thereby preserve genome stability.

## Results

### NAIL-seq identifies early DNA replication initiation zones

To identify replication initiation zones with the aim of tracing early DNA replication, we synchronized human GM12878 (primary-like) and K562 (leukemia) cells separately in the G1 phase using the CDK4/6 inhibitor palbociclib [38]. More than 94% of the treated cells were arrested in the G1 phase after treatment for 36 h (Fig. 1a; Additional file 1: Figure S1a). After a release of 2–3 h from the G1 phase, the synchronized K562 and GM12878 cells reached the G1/S transition (Additional file 1: Figure S1a and b). The K562 and GM12878 cells were labeled after a release of 2.5 or 3 h, respectively, with EdU and then BrdU, each for a 15-min pulse (Fig. 1a; Additional file 1: Figure S1a and b). Theoretically, the two thymidine analog signals can mark replication initiation sites and adjacent fork elongation regions for tens of kilobases. The EdU- and BrdU-incorporated fragments can be distinguished by using streptavidin C1 beads after the Click reaction or an anti-BrdU antibody, respectively (Additional file 1: Figure S1c). Of note, EdU cross-reacted with the anti-BrdU antibody; nevertheless, this cross-reaction was undetectable after the Click reaction (Additional file 1: Figure S1d). Incorporation of EdU or BrdU from the sequential dual-labeling revealed strong early DNA replication signals in the early replication domains of synchronized K562 cells (Fig. 1b; Additional file 1: Figure S2a). Moreover, the "E-B" signal obtained by subtracting the BrdU signal from the EdU signal was further narrowed to the middle of the early replication domains (Fig. 1b; Additional file 1: Figure S2a). The resolution of the E-B peaks was more than 2-fold higher than that of the individual EdU and BrdU peaks, with a median width of 90 kb versus 245 and 290 kb, respectively, in K562 cells (Fig. 1c; Additional file 1: Figure S2b).

Next, we employed a second assay, EdU-seq-HU, by slowly incorporating EdU under HU treatment to map early replication initiation [13, 15]. G1-arrested cells were released into fresh medium containing HU and EdU for an additional 12 h prior to harvesting (Fig. 1a; Additional file 1: Figure S1c). Approximately 71.8% of EdU/HU hotspots in K562 cells overlapped with 78.9% of the E-B peaks (Fig. 1d). The EdU/HU hotspots showed a median width of 55 kb, exhibiting a slightly higher resolution than that of the E-B peaks (Fig. 1c; Additional file 1: Figure S2b). HU treatment yields high-resolution replication-associated peaks [13, 15], but it may induce DNA double-stranded breaks (DSBs) and early utilization of dormant DNA replication origins [39–42]. E-B signals are obtained from cells undergoing the G1/S transition without replication stress; therefore, E-B signals can be used to exclude early firing of dormant origins and potential DNA damage in the EdU/HU libraries. The EdU/HU peaks overlapping the E-B peaks showed earlier replication timing in comparison with that of the non-overlapping EdU/HU peaks (Fig. 1e). Therefore, we defined the EdU/HU peaks overlapping the E-B peaks as ERIZs and subjected them to further analysis (Fig. 1a, b, and d; Additional file 1: Figure S2a). Since EdU-seq-HU also employed a nucleoside analog for

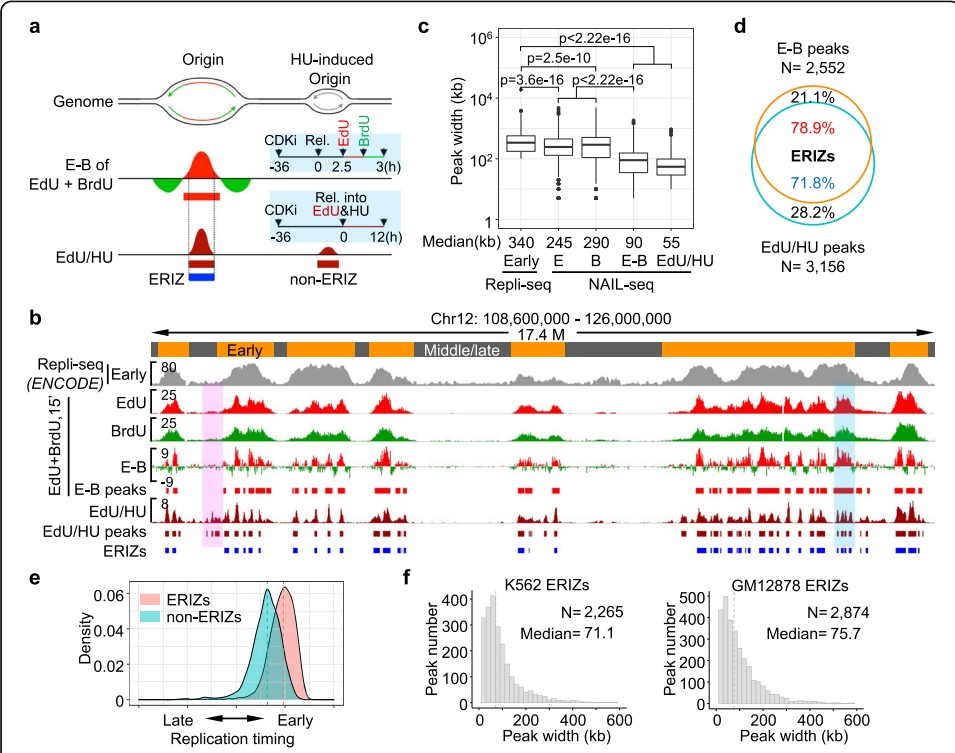

**Fig. 1** NAIL-seq identified DNA early replication initiation zones (ERIZs). **a** Schematic showing the ERIZs identified by NAIL-seq. Nascent DNA labeled with EdU is red, while that labeled with BrdU is green. The origin indicated by the gray arrows was identified by EdU/HU, but not by EdU and BrdU. The light blue boxes show the procedures for the indicated NAIL-seq libraries. CDKi is the CDK4/6 inhibitor, palbociclib. Rel., release; ERIZ, early replication initiation zone. **b** Early replication signals from NAIL-seq libraries in K562 cells. Bar plots show the distribution of EdU (red) or BrdU (green) signals in the indicated region. The early DNA replication regions identified by Repli-seq are shown in gray. For the dual-labeled samples, the G1-arrested K562 cells were released for 2.5 h to reach the G1/S transition and then sequentially labeled with EdU and BrdU, each for 15 min. The E-B panel shows the BrdU-subtracted EdU signals; red represents the EdU-dominant signal and green represents the BrdU-dominant signal. E-B peaks called by the RepFind pipeline are shown underneath. For EdU/HU, G1-arrested cells were released into medium supplied with 10 mM HU and 10 μM EdU for 12 h before harvesting. EdU/HU signals are shown in dark red with the identified ERIZs in blue underneath. The pink shadow box highlights a non-ERIZ region, while the cyan box highlights a typical region where EdU/HU displays a higher resolution than E-B. **c** The width distribution of NAIL-seq identified replication peaks and Repli-seq revealed early replicated regions in K562 cells. The Wilcoxon Rank-sum test was applied for statistical analysis. **d** Venn diagram showing the overlapping peaks of E-B and EdU/HU in K562 cells. The cyan circle represents the total number of E-B peaks and the orange circle represents the EdU/HU peaks. The EdU/HU peaks overlapping with E-B peaks are defined as ERIZs. **e** Replication timing analysis for the two categories of EdU/HU peaks in K562 cells. EdU/HU peaks are categorized as ERIZs (n = 2,265, red, overlapping with E-B peaks) and non-ERIZs (n = 891, black, independent of E-B peaks). The replication timing of each peak is defined by the mean of the wavelet-smoothed signals from six fractions of the ENCODE Repli-seq profile. **f** The peak width distribution of ERIZs in K562 and GM12878 cells. The total number (N) and the median width of ERIZs are shown in the legends

sequencing, this method was termed nucleoside analog incorporation loci sequencing (NAIL-seq). NAIL-seq identified 2265 and 2874 ERIZs with a median size of 71 kb and 76 kb in K562 and GM12878 cells, respectively (Fig. 1f; Additional file 2: Table S1; Additional file 3: Table S2). As expected, more than 95% of the identified ERIZs were found in early replication regions in K562 and GM12878 cells (Additional file 1: Figure S2c). Of note, only 0.04% (1 of 2265) of ERIZs in K562 cells and 1% (30 of 2874) of those in GM12878 cells occurred in typical late replication regions.

## NAIL-seq has higher resolution than Repli-seq

To determine whether NAIL-seq is superior to typical single-signal incorporation assays, we used published Repli-seq data from asynchronized K562 cells to perform a direct comparison of the methods [19]. The individual EdU and BrdU peaks from NAIL-seq had a markedly higher resolution in comparison with that of the early replication peaks identified by Repli-seq, which had a median peak width of 340 kb (Fig. 1b, c; Additional file 1: Figure S2a), suggesting that cell synchronization improved the resolution of the replication initiation zones identified by the experiments. With regard to ERIZs, the narrowest ERIZs identified by NAIL-seq were approximately 10 kb in size, while the narrowest early replication domains from the Repli-seq results were more than 100 kb in size (Fig. 1c, f). OK-seq and SNS-seq have also been widely used to identify initiation zones (IZs) and replication origins, respectively, associated with replication events throughout the entire S phase in unsynchronized cells. In K562 cells, nearly 50% of ERIZs identified by NAIL-seq overlapped with 43.4% of the IZs identified by OK-seq [24], and nearly 70% of these ERIZs overlapped with 16.3% of the replication origins identified by SNS-seq [23] (Additional file 1: Figure S2d). Of note, we also found that two or more replication origins identified by SNS-seq fell into a single ERIZ (highlighted in cyan in the Additional file 1: Figure S2a). Moreover, we performed EdU-seq-HU in stimulated mouse splenic B cells and the identified early replication zones were highly correlated with previous results [15] (Additional file 1: Figure S2e).

## Early DNA replication is present in non-transcribed regions

Next, we sought to investigate the relationship between transcription and early replication initiation. Roughly 94–97% of the clustered ERIZs were present in transcription-occupied A compartments [1] in K562 and GM12878 cells (Fig. 2a), consistent with previous reports [43]. To further characterize the chromatin localization of ERIZs, we built a logistic regression model to demonstrate the correlation between ERIZs and histone modification markers, as well as chromatin structural proteins, based on ChIP-seq data from ENCODE. The most significant predictive marker for ERIZs was H2A.Z (Fig. 2b; Additional file 1: Figure S3a), in line with a recent report that H2A.Z recruits ORC1 to initiate DNA replication [44]. Cohesin, CTCF, and other histone markers for transcription enhancers or promoters, including H3K4me1, H3K4me3, and H3K27ac, also had positive prediction scores for ERIZs (Fig. 2b; Additional file 1: Figure S3a). However, transcription elongation markers H3K79me2 and H3K36me3, as well as transcription silencer marker H3K9me3, had negative prediction scores for ERIZs (Fig. 2b).

To further explore the relationship between transcription and early DNA replication, we examined the zoomed-in profile of global run-on (GRO)-seq transcription signals and ERIZ-associated early replication signals. Remarkably, we found that ERIZs are mainly located in non-transcribed regions, mutually exclusive with transcribed gene bodies, despite being in the same A compartments (Fig. 2c). The genome-wide analysis confirmed that ERIZ signals accumulated in non-transcribed regions flanked by transcription signals of GRO-seq [45] in GM12878 and K562 cells (Fig. 2d). We also identified ERIZs from two primary cell types, mouse embryonic stem cells (mESCs) and mouse splenic B cells (mSBCs), and obtained similar findings (Additional file 1: Figure S3b). Of note, the mESCs were arrested by thymidine and then nocodazole [46], after

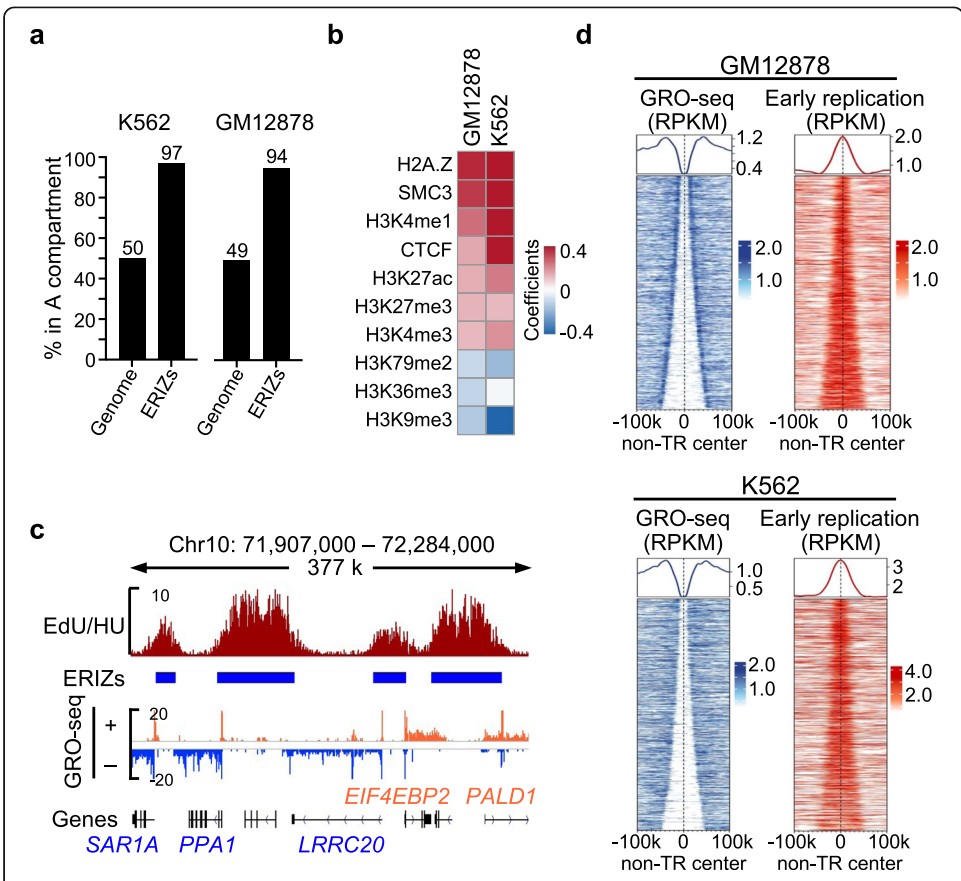

**Fig. 2** ERIZs are located in non-transcribed regions within active compartments. **a** Percentage of the width of ERIZs in the A compartments. Genome represents the percentage of A compartments in the hg19 genome in the indicated cells. **b** The relationship between the peaks of predictors and ERIZ appearance in GM12878 and K562 cells (see "Methods" for details). **c** Distribution of ERIZs in the context of active transcription in GM12878 cells. "+" indicates the forward strand and "−" indicates the reverse strand. The positions and transcribed direction of genes are shown at the bottom of the panel with the gene names. Forward-transcribed genes are marked in orange and reverse-transcribed genes are marked in blue. **d** Heatmaps of ERIZ-associated early replication initiation (from EdU/HU, red) and active transcription (blue) in GM12878 and K562 cells. The ERIZ-occupied non-transcribed regions (non-TRs) are ranked by width and centered on the midpoint flanked by two transcribed genes in each cell line. The non-TRs in A compartments, 20–100 kb in width, are displayed. Each line represents a non-TR

which they were released into medium containing HU for 3 h to allow the cells to enter the early S phase before harvesting (Additional file 1: Figure S3c and d). The G0-phase mSBCs were activated by cytokines in medium containing HU for 28 h to allow the cells to enter the early S phase [15] before harvesting (Additional file 1: Figure S3c). These results explain the negative correlation between ERIZs and transcription elongation markers and suggest that early DNA replication initiation occurs preferentially at non-transcribed regions within active chromatin compartments.

## Transcription shapes early DNA replication initiation

To investigate whether transcription interacts with DNA replication initiation, we used α-amanitin to inhibit transcription [47] in synchronized K562 cells, after which we measured the impact of transcription perturbation on the distribution of ERIZs. We

released G1-arrested K562 cells into fresh medium containing 2 or 10 μg/mL α-amanitin, in addition to HU and EdU, for 12 h before harvesting (Additional file 1: Figure S4a). Transcription blockade by α-amanitin slightly reduced EdU incorporation, but robust EdU signals were detected without significant elevation in DNA damage marked by γ-H2AX (Additional file 1: Figure S4b and c). Enrichment of early replication signals in ERIZ-associated non-transcribed regions (as defined in Fig. 2d) decreased as the α-amanitin concentration was increased (Fig. 3a; Additional file 1: Figure S4d). Early replication signals also showed increased enrichment at transcription start sites (TSSs) following α-amanitin treatment (Figs. 3a and 4f; Additional file 1: Figure S4d), possibly because TSS regions where ORC is located support DNA replication initiation, as previously reported [10, 12, 23, 48]. The ratio of early replication signals in non-transcribed regions to that of early replication signals in adjacent transcribed regions significantly decreased as the concentration of α-amanitin was increased, suggesting that early replication penetrated transcribed regions (Fig. 3b; Additional file 1: Figure S4d). Moreover, stronger inhibition of transcription resulted in more dramatic early replication redistribution on the genome (Fig. 3b and exemplified in Additional file 1: Figure S4d). Similarly, early replication signals also relocated from non-transcribed regions to neighboring transcribed regions in low-dose α-amanitin-treated

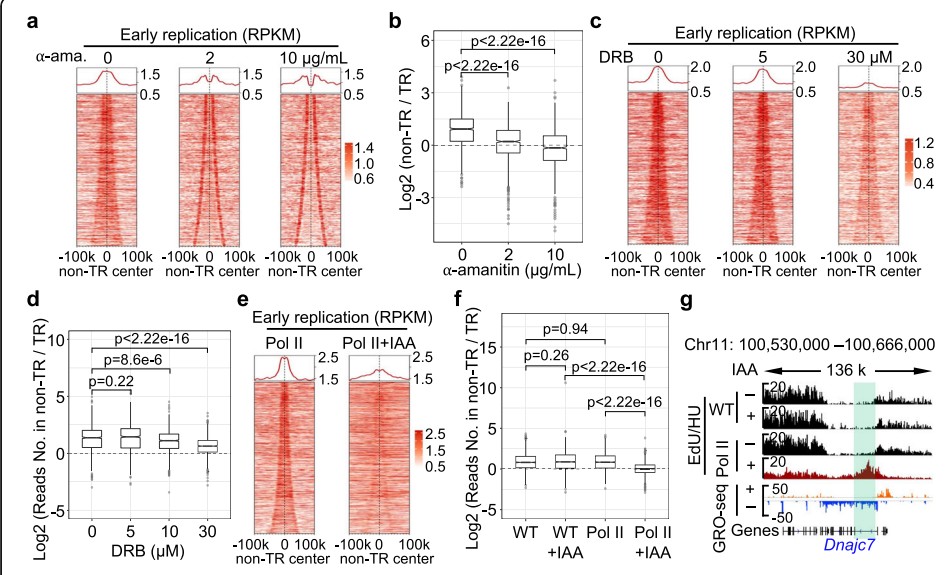

**Fig. 3** Transcription relocates early replication in K562 cells and mESCs. **a**, **c** Heatmaps of early replication signals (from EdU/HU) around ERIZ-associated non-TRs in K562 cells treated with the indicated concentrations of α-amanitin (**a**) or DRB (**c**). Cells were treated as shown in Additional file 1: Figure S4a. Non-TR, non-transcribed region. TR, transcribed region. Legends are depicted as described in Fig. 2d. **b**, **d** Box-plots showing the log2 ratio of read density between early replication in the ERIZ-associated non-TRs and that in the flanked transcribed regions in the A compartments of K562 cells treated with the indicated concentrations of α-amanitin (**b**) or DRB (**d**). The Wilcoxon rank-sum test was applied for statistical analysis. **e** Heatmaps of early replication signals (from EdU/HU) around the ERIZ-associated non-TRs in mESCs with or without RNA polymerase II. Cells were treated as illustrated in Additional file 1: Figure S4h. Endogenous RNA polymerase II tagged with mAID is marked as Pol II. Non-transcribed regions (non-TRs) are ranked by width and centered on the midpoint flanked by two transcribed genes in mESCs. The non-TRs within A compartments, 20–100 kb in width, are displayed. Each line represents a non-TR. **f** The log2 ratio of read density between early replication in ERIZ-associated non-TRs and that in the flanked transcribed regions within the A compartments of mESCs. WT, wild-type. The Wilcoxon rank-sum test was employed for statistical analysis. **g** Distribution pattern of early replication initiation with or without RNA polymerase II in mESCs

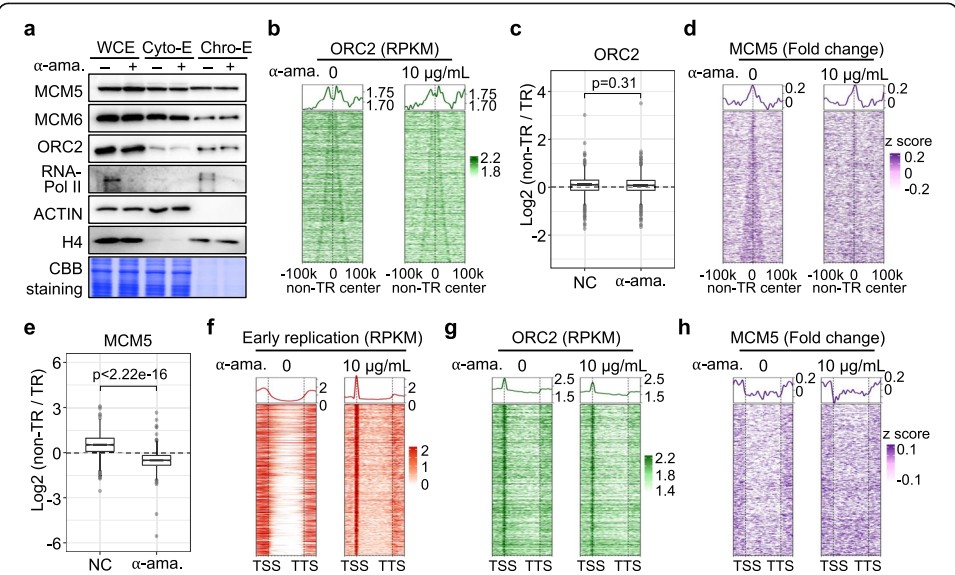

**Fig. 4** Transcription redistributes MCM in the G1 phase. **a** Western blotting showing detection of the indicated proteins in G1-arrested K562 cells in the absence or presence of 10 μg/mL α-amanitin. G1-arrested K562 cells were treated with or without 10 μg/mL α-amanitin for 12 h before harvest. WCE, whole cell extract; Cyto-E, cytoplasmic extract; Chro-E, chromatin extract. **b** The distribution of ORC2 within the ERIZ-associated non-transcribed regions (non-TRs) in G1-arrested K562 cells in the absence or presence of 10 μg/mL α-amanitin. The non-TRs are the same as those shown in Fig. 3a. **c** Box-plot showing the log2 ratio of the read density of ORC2 between non-TRs and flanked TRs in the A compartments of K562 cells treated with 10 μg/mL α-amanitin. The Wilcoxon rank-sum test was applied for statistical analysis ($p$ = 0.31). **d** The distribution of MCM5 in G1-arrested K562 cells in the absence or presence of 10 μg/mL α-amanitin. The data from ChIP-ed samples over the input samples are defined as fold change data, as in the *ENCODE* project and a previous report [35]. The fold change data were normalized to z-scores for heatmaps. The displayed regions are the same as in **b**. **e** Box-plot showing the log2 ratio of MCM5 signal enrichment in non-TRs and TRs in the A compartments of K562 cells treated with 10 μg/mL α-amanitin. The Wilcoxon rank-sum test was applied for statistical analysis. **f**, **g**, and **h** The distribution of early replication initiation from EdU/HU (**f**), ORC2 (**g**), and MCM5 (**h**) in ERIZ-adjacent active gene bodies with or without 10 μg/mL α-amanitin treatment. TSS, transcription start site; TTS, transcription termination site. ERIZ-flanked transcribed regions larger than 50 kb are ranked by width (with the smallest genes on top). For display, all transcribed regions were scaled to the same width and aligned at both TSS and TTS. Each line represents an individual transcribed gene

mSBCs (Additional file 1: Figure S4e and S4f). We also applied another transcription inhibitor, 5,6-dichloro-1-β-d-ribofuranosyl benzimidazole (DRB), at concentrations ranging from 5 to 30 μM to interrupt transcription elongation in G1-arrested K562 cells and obtained similar, albeit milder, findings (Fig. 3c, d; Additional file 1: Figure S4g).

Since transcription inhibitors may have non-specific effects on DNA metabolism, we employed the auxin-inducible degron system to directly eliminate RNA polymerase II from mESCs in an acute manner as previously reported [49]. Briefly, the catalytic subunit of RNA polymerase II was tagged with a mAID tag, and indole-3-acetic acid (IAA) was added to induce degradation of the fusion protein. mESCs were arrested at the M phase and then released into the early S phase (Additional file 1: Figure S4h). In order to eliminate RNA polymerase II before the G1/S transition, IAA was added 1 h before release from the M phase. RNA polymerase II was degraded almost completely after 2 h of induction by IAA, while the number of MCM complexes remained constant (Additional file 1: Figure S4h and i). In addition, rapid degradation of RNA polymerase II did not influence the entry of mESCs into the S phase (Additional file 1: Figuer S4j).

IAA treatment also did not significantly change the distribution of ERIZ-associated early replication signals in wild-type cells (Fig. 3f; Additional file 1: Figure S4k; Additional file 4: Table S3). However, the absence of RNA polymerase II resulted in a remarkable decrease in the number of early replication signals in non-transcribed regions (Fig. 3e), in line with the findings obtained using transcription inhibitors in K562 cells. In addition, the number of early replication signals was declined in non-transcribed regions and increased in gene bodies, as exemplified in the *Dnajc7* gene (Fig. 3f, g). Collectively, our data indicate that transcription actively orchestrates early DNA replication initiation in both human and mouse cells.

### Transcription redistributes the MCM, but not the ORC, to non-transcribed regions

It has been reported that the ORC is distributed widely in both transcribed and non-transcribed regions, with enrichment at TSSs, and this finding was confirmed by re-analysis of published data from an ORC2 ChIP-seq study [31] in asynchronized K562 cells (Additional file 1: Figure S5a). In contrast, MCM complexes accumulate in non-transcribed regions, as shown by the ChIP-seq data for MCM3, MCM5, and MCM7 from ENCODE in asynchronized K562 cells, indicating that MCM complexes are uncoupled from the ORC (Additional file 1: Figure S5a).

To explore the mechanism by which transcription influences early DNA replication initiation, we examined the distribution of the ORC and MCM via ChIP-seq following α-amanitin-induced transcription perturbation in the G1 phase prior to DNA replication initiation. The protein levels of the ORC and MCM remained constant in the cytosol and on chromatin despite the degradation of RNA polymerase II in G1-arrested K562 cells following treatment with 10 μg/mL α-amanitin (Fig. 4a; Additional file 1: S5b). Of note, ORC2 ChIP-seq was achieved by expressing Avi-tagged ORC2 at an expression level much lower than that of endogenous ORC2, which had no impact on chromatin-bound MCM5 in the G1 phase (Additional file 1: Figure S5c). ORC2 and MCM5 showed distinct distribution patterns in the G1 phase without α-amanitin treatment, which were similar to those of asynchronized K562 cells (Fig. 4b left and 4d left; Additional file 1: Figure S5a, d, and e). Following α-amanitin treatment, ORC2 primarily remained at its original binding sites, showing similar distribution patterns in both non-transcribed and transcribed regions regardless of α-amanitin treatment (Fig. 4b, c). However, the MCM5 signal was diminished in non-transcribed regions, and the ratio of the MCM5 signal in non-transcribed regions to that of the signal in transcribed regions decreased significantly (Fig. 4d, e).

To better present changes in early replication, ORC, and MCM signals in transcribed regions, we aligned ERIZ-adjacent transcribed genes at both TSS and transcription termination sites (TTS), which revealed that early replication signals were rarely detected within transcribed genes before treatment (Fig. 4f). Following α-amanitin treatment, early replication signals were detected in transcribed regions, with obvious enrichment at TSSs (Fig. 4f), while the abundance of ORC2 remained unchanged within transcribed genes and was enriched at TSSs (Fig. 4b, c, and g). However, the MCM5 signal within transcribed genes was intensified, and mild signal pileups were observed within gene bodies (Fig. 4e, h). In particular, α-amanitin treatment resulted in similar distribution patterns for ERIZs and ORC2 (Fig. 4f right versus 4g right). These findings imply that

RNA polymerase II redistributes MCM complexes outside of transcribed regions, driving uncoupling of the ORC and MCM complexes and thus preventing early DNA replication initiation at transcribed regions.

### MCM accumulates at RNA polymerase II blockade sites to initiate DNA replication

The driving of MCM complexes by RNA polymerase II requires the coupling of these two complexes in transcribed regions in the G1 phase. To detect the coupling of RNA polymerase II and MCM double hexamers, we utilized a transcription barrier to stall RNA polymerase II and thereby trap MCM. In this assay, trapped MCM accumulates and may induce DNA replication initiation in the arrival direction of transcription

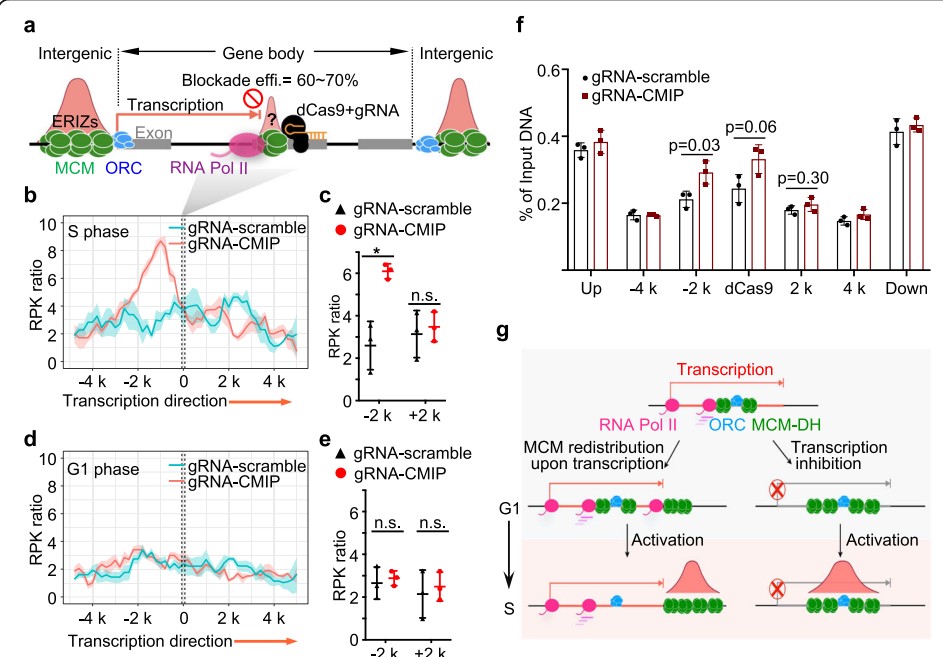

**Fig. 5** Replication initiation occurs at the transcription barrier. **a** Schematic of dCas9-gRNA-mediated transcription blockade. Four gRNAs were designed to bind the non-template strand at the fourth exon of *CMIP*. The orange arrow indicates the transcribed region and the red stop sign illustrates the dCas9-induced transcription blockade. **b**, **d** The mean of EdU/HU normalized read density (line) with the standard error (shadow) of three biological replicates near the dCas9/gRNA binding sites on *CMIP* in early S phase (**b**) or G1-phase (**d**) K562 cells treated with *CMIP* (red) or scrambled (cyan) gRNAs. gRNA-CMIP and gRNA-scramble indicate that cells were treated with dCas9/gRNA targeting *CMIP* or a control region, respectively. The regions between two dashed lines around 0 highlight the four binding sites of CMIP gRNAs. The 10-kb window of dCas9 binding sites is tiled by 1-kb bins (sliding by 200 bp). The read count per kilobase (RPK) was calculated within each 1-kb bin and normalized by the RPK of the B compartments from each biological replicate, defining the RPK ratio (see "Methods" for details). **c**, **e** The levels of EdU/HU read density upstream (− 2 k) or downstream (2 k) of the gRNA-CMIP binding sites in early S phase (**c**) or G1 phase (**e**) K562 cells. The read density was calculated in 2-kb bins. Student's *t*-test, *p* = 0.025; *, *p* < 0.05; n.s., no significance. **f** ChIP-qPCR showing the distribution of MCM5 surrounding dCas9-binding sites in *CMIP*. A biological replicate of MCM5 ChIP-qPCR and the other two biological replicates are shown in Additional file 1: Figure S6i. *t*-test, mean ± SD. "Up" or "down" indicates the upstream or downstream non-transcribed regions of *CMIP*, respectively. The other five positions are the same as those shown in **b** and **d**. **g** Schematic of the transcription bulldozing model. Left: in the G1 phase, MCM double hexamers (MCM-DHs) loaded by ORC in transcribed regions are driven along gene bodies by RNA polymerase II to downstream non-transcribed regions. Accumulated MCM-DHs in non-transcribed regions initiate DNA replication in the early S phase. Right: when transcription is inhibited, MCM-DHs accumulate around the ORC binding site and initiate DNA replication at gene bodies

upstream of the barrier. We designed four guide (g)RNA-target sites spanning a 162-bp exonic region of the *CMIP* gene without ORC binding capability to recruit nuclease-dead Cas9 (dCas9) as a transcription roadblock (Fig. 5a). The efficiency of transcription blockade by dCas9 reached 60–70%, and the upstream to downstream transcription levels were significantly increased (Additional file 1: Figure S6a-c). Both the upstream and downstream ERIZs neighboring *CMIP* still supported substantial early DNA replication following the dCas9-mediated transcription blockade as a result of the presence of the ORC in neighboring non-transcribed regions and inefficient transcription inhibition (Additional file 1: Figure S6b-d). Interestingly, we found enrichment of EdU signals within the 2-kb region upstream of the dCas9 binding sites in the arrival direction of transcription, but not in the other direction (Fig. 5a–c; Additional file 1: Figure S6e). To evaluate whether these EdU signals were produced by DNA replication initiation or DNA damage due to transcription stalling, we performed EdU-seq-HU in the G1-arrested K562 cells. We found no EdU enrichment within the same 2-kb region following transcription blockade (Fig. 5d, e; Additional file 1: Figure S6b and e). Given that both transcription and DNA replication occurred in the S phase, but only transcription occurred in the G1 phase, the enriched EdU signals in the early S phase must have originated from new DNA replication initiation. In addition, no significant early replication signals were observed within the gene body downstream of the dCas9 block site, possibly because ORC was not present (Fig. 5b, c; Additional file 1: Figure S6d and e). We also performed dCas9-induced transcription blockade at *GALNT10*. Although the blockade efficiency was lower than that observed at *CMIP* (Additional file 1: Figure S6b and c), we observed abrupt DNA replication initiation following dCas9-induced transcription blockade (Additional file 1: Figure S6f-h).

To detect MCM accumulation, we performed anti-MCM5 ChIP-qPCR on *CMIP* in the G1 phase following dCas9-induced transcription blockade, because ChIP-seq is not suitable for detecting minor changes at individual sites. dCas9-mediated stalling of RNA polymerase II did not alter the amount of MCM5 in the upstream or downstream non-transcribed regions of *CMIP* (Fig. 5f). Remarkably, MCM5 significantly accumulated in the 2-kb region upstream of the blockade sites in three biological replicates, consistent with the distribution pattern of EdU signals in the early S phase following dCas9-induced transcription blockade (Fig. 5b, c, and f; Additional file 1: Figure S6i). These findings suggest that RNA polymerase II may drive MCM along active genes to transcription-poor regions or transcription obstacles to initiate early DNA replication (Fig. 5g).

## Collisions between transcription and DNA replication initiation induce genome instability

It is conceivable that the exclusion of early DNA replication initiation from transcribed regions may represent a mechanism for preserving genome integrity, especially in transcribed regions. Therefore, we investigated the impact of collisions between dysregulated early DNA replication and transcription on genome stability. We used 30 μM DRB to interrupt transcription in G1-arrested K562 cells, which allowed DNA replication initiation to occur within actively transcribed genes (Fig. 3c, d). Subsequently, the DRB treatment was ceased for 3.5 h to restart transcription and induce transcription-replication collisions, and γ-H2AX was used to mark DNA damage (Fig. 6a). Treatment

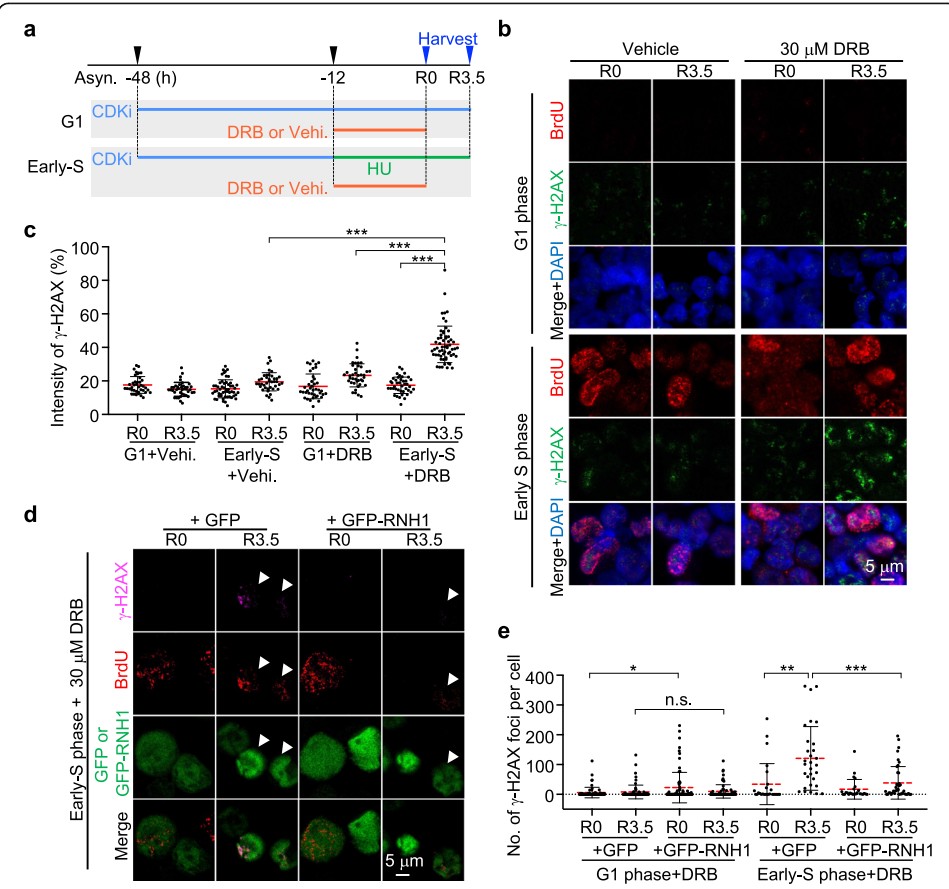

**Fig. 6** Relocated early replication initiation induces DNA damage. **a** Schematic showing DRB treatment of G1 or early-S phase K562 cells; a detailed description is given in the "Methods" section. The blue arrows show the samples that are presented in **b** and **c**. **b** Detection of nuclei-resident γ-H2AX foci in G1-arrested or early S phase K562 cells following transcription re-initiation via withdrawal of DRB. **c** Intensity of nuclei-resident γ-H2AX foci following transcription re-initiation via withdrawal of DRB in G1-arrested or early S phase K562 cells. Representative images are shown in **b**. *t*-test; ***, *p* < 0.001. **d** Detection of γ-H2AX foci induced by DRB withdrawal with or without RNase H1 (GFP-RNH1) overexpression. The white triangle indicates early S phase cells with GFP or GFP-RNH1 overexpression at 3.5 h after DRB withdrawal. Of note, the antibody against γ-H2AX is different from the one in **b**. **e** The number of nuclei-resident γ-H2AX foci with or without RNase H1 overexpression in the G1 or early S phase before or after withdrawal of DRB. *t*-test; n.s., no significance; *, *p* < 0.05; **, *p* < 0.01; ***, *p* < 0.001

with 30 μM DRB caused similar levels of damage in cells in the G1 and early S phases in comparison with untreated cells (Fig. 6b, c). After DRB treatment was ceased for 3.5 h, the intensity of the γ-H2AX signal in G1-arrested K562 cells increased slightly in comparison with that of the untreated cells (Fig. 6b, c). However, in the DRB-released early S phase cells, in which DNA replication initiation occurred in transcribed regions (Fig. 3c, d), the γ-H2AX signal intensity was significantly higher than that of the G1-phase or untreated cells (Fig. 6b, c).

R-loops are important deleterious structures produced during the collision of replication and transcription, which are sensitive to RNase H1 [8, 9]. In this context, we over-expressed RNase H1 in the DRB-treated-and-released K562 cells by plasmid transfection (Fig. 6a, d; Additional file 1: Figure S7a). Overexpression of RNase H1 caused DNA damage in asynchronized cells [50] and slightly, but significantly, in-creased the γ-H2AX signal in the G1-arrested cells before, but not after, DRB release

(Fig. 6e; Additional file 1: Figure S7a). The γ-H2AX signal was elevated after DRB release in the GFP-overexpressed K562 cells in the early S phase, consistent with the findings described above. Overexpression of RNase H1 resulted in a significant decrease in the number of γ-H2AX foci in comparison with that of the GFP-overexpressed DRB-released S phase cells (Fig. 6d, e), indicating that RNase H1 alleviated DNA damage in DRB-released cells undergoing replication-transcription collisions. Collectively, these results suggest that genome instability is increased when transcription is restarted within transcribed regions with dysregulated DNA replication initiation.

## Discussion

It is well established that thymidine analogs can be incorporated into the genome to map DNA replication loci. CldU and IdU have been used in DNA combing assays to analyze DNA replication events at a single-molecule level using two distinct antibodies [51]. Here, we used two different enriching strategies to detect sequentially incorporated EdU and BrdU signals and thus map replication loci via high-throughput sequencing. To ensure that EdU and BrdU-labeled replication initiation zones and adjacent elongation regions, cells were synchronized in the G1 phase using a CDK4/6 inhibitor, released at the G1/S transition stage, and sequentially labeled by EdU and BrdU for short pulses. Since no extra replication stress is present, NAIL-seq with dual-labeling of EdU and BrdU can more accurately identify early replication initiation in comparison with EdU-seq-HU (Fig. 1b, e). Moreover, NAIL-seq has a much higher resolution than Repli-seq, even when S phase cells are divided into 16 fractions for Repli-seq library generation [20]. However, NAIL-seq with dual-labeling has a slightly lower resolution than EdU-seq-HU (Fig. 1c; Additional file 1: Figure S2b), and library preparation for the dual-labeling assay is more time-consuming. Ultimately, we combined dual-labeling of EdU and BrdU with EdU-seq-HU to precisely locate bona fide ERIZs in this study.

   Both transcription and early DNA replication occur in open chromatin compartments, and each of these two pivotal cellular processes is indispensable to the other [1, 5, 7, 8]. Transcription plays a dual role in regulating early DNA replication. On the one hand, high-level transcription is coincident with robust DNA replication initiation in the same active compartments, and enhanced transcription can induce late replicating regions to replicate earlier, indicating that transcription and DNA replication initiation are collaborative [12, 14, 21, 23, 48, 52]. On the other hand, transcription can inhibit replication initiation at origins, as reported in budding yeast [53–55]. In this study, we propose a "transcription bulldozing" model to describe the RNA Pol II-driven MCM redistribution mechanism that spatially separates transcription and early DNA replication (Fig. 5g). We concluded that RNA polymerase II, analogous to a bulldozer in this model, pushes away MCM to suppress DNA replication initiation in transcribed regions (Fig. 4). Our results explain the observation that early DNA replication initiation is rarely detected at transcribed genes [10, 15]. According to the "transcription bulldozing" model, DNA replication initiation is predisposed to start at transcription barriers or other chromatin structural obstacles such as TSSs, G-quadruplex stretches, Z-DNA, and hairpins [10, 21–23, 48, 56–58]. Consistently, ERIZs are also enriched in native transcription barriers including G-quadruplex stretches (Additional file 1: Figure S7b). Here, we introduced a dCas9-induced transcription blockade at the transcribed regions of *CMIP* and *GALNT10*, providing another piece of evidence suggesting that RNA

polymerase II may drive MCM complexes along transcribed genes (Fig. 5). When encountering the dCas9 blockade, RNA polymerase II-coupled MCM complexes were trapped upstream of dCas9-binding sites. As expected, we detected an early replication signal correlated with the trapped MCM complexes at the block (Fig. 5; Additional file 1: Figure S6). Regarding regions downstream from the block, loaded MCM double hexamers would not be expected to be cleared and might also initiate DNA replication. However, the lack of the ORC at the *CMIP* gene downstream of dCas9-binding sites resulted in the absence of MCM complexes and thereby DNA replication initiation (Fig. 5b; Additional file 1: Figure S6b-e).

The MCM complex experiences two phases, loading and activation, when DNA replication is initiated in eukaryotic cells [26]. In the loading step, which occurs during the late M/G1 phase, multiple head-to-head MCM double hexamers are loaded onto chromatin by the ORC at each replication origin [59]. These loaded MCM double hexamers are inactive and encircle the duplex DNA, allowing them to move after loading [29, 30, 35, 36]. In this context, we detected uncoupling of the ORC and MCM complexes in both cycling and G1-arrested human cells (Fig. 4b, d; Additional file 1: Figure S5a), consistent with previous reports [60, 61]. Since activated MCM double hexamers, but not the ORC, determine whether DNA replication initiation loci are functional [32, 62, 63], RNA polymerase II can effectively shape early DNA replication initiation by redistributing MCM double hexamers prior to activation. In addition, active transcription shapes the genome-wide distribution of MCM complexes in *Drosophila* [35]. Moreover, in vitro studies demonstrated that MCM double-hexamers from *Saccharomyces cerevisiae* can slide on naked duplex DNA [29, 30], and T7 RNA polymerase can push MCM complexes off of the ends of linear dsDNA [36] via a process supported by studies of MCM-DNA structure [64, 65]. In the context of chromatin, in vivo ChIP-seq also showed that MCM complexes were relocated from their loading sites to adjacent downstream regions along the transcription direction upon transcription read-through in budding yeast [36, 37]. In this study, we found that genome-wide redistribution of MCM complexes was shaped by RNA polymerase II-dependent transcription in mammalian cells (Fig. 4). Moreover, RNA polymerase II can relocate MCM complexes to dCas9-binding sites within active gene bodies (Fig. 5), supporting the "transcription bulldozing" model. Furthermore, we observed that the abundance of MCM complexes on chromatin was comparable before and after transcription perturbation (Fig. 4a), implying that RNA polymerase II may push MCM complexes along chromatin rather than disassociating them from transcribed regions, in line with the observation that T7 RNA polymerase cannot disassociate budding yeast MCM complexes from circular DNA in vitro [36]. However, we cannot fully exclude the possibility that RNA polymerase II may disassociate MCM complexes from chromatin in the G1 phase, as it has been reported that transcription can disrupt pre-RCs in budding yeast [53–55]. The mechanism by which RNA polymerase II redistributes MCM complexes on chromatin remains to be explored.

During the S phase, DNA replication initiation sites are prone to DNA damage and transcription is robust [9, 15, 39, 66]. Therefore, suppression of DNA replication initiation within transcribed regions is important because it allows cells to avoid collisions between DNA and RNA polymerases, thus maintaining genome stability during early DNA replication. Accordingly, we detected an increased level of genome instability

when the transcription machinery encountered induced DNA replication initiation at gene bodies (Fig. 6a–c). Moreover, inducing a disordered cell cycle via overexpression of oncogenes (c-MYC or cyclin E) results in DNA replication initiation in gene bodies, which promotes chromosomal translocation and tumorigenesis [13]. In this context, employing RNA polymerase II to exclude inactive MCM double hexamers from active gene bodies is a superbly effective strategy by which cells initiate DNA replication while maintaining robust gene transcription.

## Conclusions

Both early DNA replication initiation and transcription occur in active chromatin compartments; however, the mechanisms through which these pivotal cellular processes coordinate remain unclear, especially in mammalian cells. Here, we report our development of the NAIL-seq method to monitor the interplay between early replication initiation and transcription, which allowed us to reveal that transcription actively shapes early DNA replication initiation in mammalian cells. Moreover, we describe the mechanism through which RNA polymerase II actively redistributes replication initiation factor MCM complexes to non-transcribed regions to avoid vulnerable early DNA replication initiation within transcribed genes. We also report that inducing collisions between transcription and DNA replication initiation within gene bodies leads to gross DNA damage. Therefore, we propose a "transcription bulldozing" model to describe the key role of transcription in preserving genomic stability during DNA replication initiation in mammalian cells.

## Methods

### Materials and resource table

See the Additional file 5: Table S4.

### Cell culture, cell cycle synchronization, and incorporation of thymidine analogs for NAIL-seq

GM12878 and K562 (3111C0001CCC000039, National Infrastructure of Cell line resource, China) cells were cultured in RPMI1640 media supplied with 15% FBS as described previously [1, 67]. For G1 arrest, GM12878 or K562 cells were incubated with 1 μM or 5 μM palbociclib (SelleckChem) for 36 h, respectively. For EdU and BrdU dual-labeling experiments, G1-arrested GM12878 and K562 cells were washed with pre-warmed RPMI1640 and released into fresh medium for 3 or 2.5 h, respectively. Next, the cells were cultured with 10 μM EdU for 15 min, followed by a wash with pre-warmed RPMI1640, and then cultured with 50 μM BrdU for an additional 15 min. For EdU/HU treatment, G1-synchronized GM12878 or K562 cells were released into fresh medium with 10 μM EdU plus 5 mM HU for 24 h or 10 μM EdU plus 10 mM HU 12 h, respectively. The concentration of EdU was comparable with that of endogenous dTTP [68]. EdU is efficiently incorporated into nascent DNA, and a 2-min period is sufficient for EdU to label nascent Okazaki fragments in OK-seq [10]. Moreover, G1-released K562 cells require an additional 2.5 h to reach the G1/S transition. Therefore, the vast majority of initiation sites are subjected to EdU labeling in the EdU/HU assay. For α-amanitin or DRB mediated-RNA polymerase II inhibition, G1-arrested K562 cells

were released into fresh media containing the indicated concentrations of α-amanitin or DRB with HU and EdU for 12 h.

Murine embryonic stem V6.5 (mES-V6.5) cells were grown in 2i media containing 15% FBS, 1 μM PD0325901 (SelleckChem), 3 μM CHIR99021 (SelleckChem), 1000 U/mL mouse LIF (Millipore), and other essential supplements as described previously [69]. mES-V6.5 cells were cultured on 0.2% gelatin (Sigma)-treated plates and synchronized by thymidine and nocodazole as described previously [46]. Mitotically arrested cells were shaken off, washed with pre-warmed 2i media, and cultured in fresh media for 1.5 h, followed by sequential labeling with 10 μM EdU and 50 μM BrdU for 10 min each. To reduce replication fork progression speed, M phase-synchronized mES cells were released into fresh media and incubated with 4 mM HU and 50 μM EdU for 3 h. To induce degradation of RNA polymerase II, homozygous mAID-tagged RNA polymerase II cells were first synchronized as wild-type mES cells. Next, 1 μg/mL doxycycline (Dox) was added to induce production of OsTIR1. In total, 500 μM IAA was added 1 h before the cells were released from the nocodazole treatment to induce degradation of RNA polymerase II. The nocodazole-arrested cells were then cultured in the presence of Dox, IAA, EdU, and HU for 3 h before harvesting.

Wild-type C57BL/6NCr mice (6–8 weeks old) were subjected to splenic B cell purification by following the manufacturer's instructions included with the EasySep Mouse B cell isolation kit (STEMCELL). Both male and female mice were indiscriminately used in this study. Mice were housed and handled according to the standards set by the Peking University laboratory animal center, and all animal experiments were approved by the institutional animal care and use committee at Peking University. Naïve splenic B cells were isolated and activated with LPS (25 mg/mL; Sigma) and IL-4 (5 ng/mL; Sigma). B cells were treated with 10 mM HU and 10 μM EdU for 28 h beginning from the activation [15]. For α-amanitin treatment, primary B cells were incubated with 10 mM HU, 10 μM EdU, and α-amanitin for 28 h after LPS and IL-4 were added.

### NAIL-seq

Cells were harvested, washed with PBS, and subjected to genomic DNA (gDNA) extraction with Proteinase K digestion as described previously [70]. Purified genomic DNA was subjected to the Click reaction with 100 μM Biotin-PEG-azide, 2 mM $CuSO_4$, 4 mM THPTA, and 10 mM sodium ascorbate at 25 °C for 1 h, followed by DNA precipitation. The copper in the Click reaction hydrolyzes gDNA and fragments it into small fragments (around 200~400 bp in length), so that each fragmented nascent DNA has little chance to contain both EdU and BrdU simultaneously. The dissolved DNA was denatured at 95 °C for 5 min and phosphorylated by T4 PNK at 37 °C for 30 min.

For EdU- and BrdU-labeled samples, the denatured and modified DNA was purified by anti-BrdU (BU1/75) and Protein G beads to isolate the BrdU-labeled portion, and the supernatants were denatured again and incubated with Streptavidin C1 beads (Dynabeads) to isolate the EdU-labeled portion. For EdU/HU-labeled samples, T4 PNK-treated DNA was incubated with C1 beads alone. All of the enrichment steps were performed at room temperature for 4 h.

Isolated EdU- or BrdU-labeled ssDNA samples were ligated onto the beads with two types of bridge adapters (Bridge adapter -1 and -2, see Additional file 5: Table S4 for

sequence details) simultaneously at room temperature for at least 4 h. The ligation reaction volume contained 1 μM of each Bridge adapter, 1,600 U T4 DNA ligase and 15% PEG-8000. Next, the ligated DNA was carefully washed and tagged with Illumina P5-I5 and P7-I7 sequences as described elsewhere [70]. All libraries were subjected to 2 × 150 bp Hiseq sequencing.

### ORC2 and MCM5 ChIP-seq

For the ORC2 ChIP-seq, ORC2 tagged with Avi-tag [71] at its N-terminal was inserted into the pMAX-GFP (Lonza) backbone to replace GFP. Bacterial BirA [71] was inserted into pX330 to replace Cas9. Ten million K562 cells transfected with 30 μg BirA and 30 μg Avi-ORC2 plasmids were immediately arrested at the G1 phase by 5 μM palbociclib for 36 h and then incubated with or without 10 μg/mL α-amanitin for 12 h in the presence of palbociclib. For MCM5, wild-type K562 cells were treated with palbociclib for 36 h and then cultured in the presence or absence of 10 μg/mL α-amanitin for 12 h with palbociclib.

Cells were fixed by 1% formaldehyde (F1635, Sigma) for 10 min at room temperature and quenched by 125 mM glycine for 5 min. Ten million fixed cells were washed twice with PBS and lysed with ice-cold NP40 lysis buffer (10 mM Tris-HCl, pH 7.5; 150 mM NaCl; 0.05% NP40) on ice for 15 min. The cell lysate was separated by 24% (w/v) sucrose in NP40 lysis buffer. The nuclei pellet was washed with 1 mM EDTA/PBS, resuspended in glycerol buffer (20 mM Tris-HCl, pH 8.0; 75 mM NaCl; 0.5 mM EDTA; 0.85 mM DTT; 50% glycerol), and lysed using nuclei lysis buffer (10 mM HEPES, pH 7.6; 1 mM DTT; 7.5 mM MgCl$_2$; 0.2 mM EDTA; 0.3 M NaCl; 1 M urea; 1% NP40). Chromatin pellets were washed twice with 1 mM EDTA/PBS and resuspended in sonication buffer (20 mM Tris-HCl, pH 8.0; 150 mM NaCl; 2 mM EDTA; 0.1% SDS; 1% Triton X-100; 4 mM CaCl$_2$) containing 40 U MNase for 15 min at 37 °C. The nuclease was quenched by adding 5 mM EDTA and 5 mM EGTA. The chromatin in the samples was further fragmented by sonication and separated from the soluble fraction.

For ORC2 ChIP-seq, the supernatants were pre-cleared using 40 μL protein G dynabeads for 1 h and then incubated with 40 μL Streptavidin T1 beads (Dynabeads) for 6 h at 4 °C. The beads were carefully washed with freshly made 2% SDS, high-salt buffer (500 mM NaCl; 0.1% SDS; 1% Triton X-100; 2 mM EDTA; 20 mM Tris-HCl, pH 8.0), LiCl buffer (0.25 M LiCl; 1% NP-40; 10 mM Tris-HCl; pH 8.0, 1 mM EDTA), and TE buffer. The bead-bound chromatin was eluted with elution buffer (20 mM Tris-HCl, pH 8.0; 10 mM EDTA; 1% SDS) at 65 °C overnight. For MCM5 ChIP-seq, the soluble fractions were directly subjected to anti-MCM5 (1:50) enrichment at 4 °C overnight. Next, protein G dynabeads were added and the mixture was incubated for 1 h. The beads were washed using the same procedure used for the ORC2/5 ChIP, except that the wash with 2% SDS was not performed. The chromatin on the beads was eluted twice with elution buffer at 65 °C for 15 min.

The eluate fraction was digested by RNase A and incubated with Proteinase K overnight. DNA was purified, end-repaired, and subjected to poly-dC tailing as described previously [72]. After poly-dC tailing, DNA was tagged with a biotinylated primer, captured by Streptavidin C1 beads, and ligated with a selected bridge adapter (see Additional file 5: Table S4) overnight at room temperature. The ligated products were then

washed and tagged with Illumina P5-I5 and P7-I7 sequences via PCR for 13 cycles. Sequencing was performed on the Illumina Hiseq platform (2 × 150 bp).

### dCas9-mediated transcription elongation blockade

Four gRNAs (see Additional file 5: Table S4 for sequence details) binding to the non-template strand of the fourth exon of *CMIP* were designed in a 134-bp region and inserted into a single plasmid expressing nuclease-dead *Sp*Cas9 (dCas9) following the procedures for CRISPRi [73]. One million K562 cells transfected with 5 μg plasmids were arrested at the G1 phase by 5 μM palbociclib for 36 h. The G1-arrested cells were either released into the early S phase or grown through the G1 phase with a further incubation with HU and EdU for 12 h. The cells were harvested, after which early DNA replication initiation and the transcription level of the *CMIP* gene were measured.

For MCM5 ChIP-qPCR, 10 million K562 cells were transfected with the indicated plasmids, incubated with 5 μM palbociclib immediately, and cultured for 48 h. The G1-arrested cells were harvested and ChIP-ed as described in the ChIP-seq section, except that the MNase treatment was not performed. The ChIP eluate fractions were subjected to RNA removal, decrosslinking, and purification. Next, the resuspended DNA was subjected to qPCR detection with the indicated primer sets (see Additional file 5: Table S4 for sequence details).

To block the elongation of RNA polymerase II on *GALNT10*, eight gRNAs (see Additional file 5: Table S4 for sequence details) targeting the second exon of *GALNT10* were designed in a 160-bp region and cloned into two plasmids expressing dCas9. The two plasmids were transfected into K562 cells, simultaneously, and transfected cells were treated as described above. Next, the cells were harvested, after which early DNA replication initiation and the transcription level of *GALNT10* were measured.

### Flow cytometry analysis

GM12878 or K562 cells treated with or without palbociclib were labeled with 50 μM BrdU for 30 min and fixed by 4% PFA for 1 h at 4 °C. The BrdU pulse-labeled cells were denatured by 3 M HCl, incubated with anti-BrdU (× 100, BD) for 40 min, and then stained with 7-AAD (× 250, BD) for 20 min. For EdU/HU-treated samples, EdU-labeled cells were subjected to the Click reaction following the manufacturer's instructions (Click-iT EdU Alexa Flour 488 Flow Cytometry Assay Kit). Samples were acquired on a BD FACSVerse and analyzed with FlowJo 10.4.

### Immunofluorescence

Cells were labeled with EdU or BrdU as described in the aforementioned sections and fixed with 4% PFA for 15 min, followed by a wash in PBS. Cells were denatured using 3 M HCl and neutralized in 0.1 M $Na_2B_4O_7$. To detect the cross-reaction between anti-BrdU and EdU (Additional file 1: Figure S1d), denatured cells were Click-reacted with TAMRA-azide-biotin or DMSO, blocked with BSA, and incubated with anti-BrdU (BU1/75), followed by incubation with the indicated secondary antibody. For cell cycle progression detection (Additional file 1: Figure S1b), denatured samples were blocked with 5% BSA, incubated with an antibody against BrdU (BU1/75) for 1 h followed by a thorough wash, and then incubated with the indicated secondary antibody. For γ-

H2AX detection (Additional file 1: Figure S4c), G1-arrested K562 cells were released into fresh medium containing 10 mM HU plus 10 μM EdU with or without amanitin treatment for 12 h. The fixed cells were Click-reacted with Azide-Alexa-555 and then incubated with antibodies against γ-H2AX and RNA polymerase II simultaneously. For the early S phase samples used in the DRB removal assay, G1-arrested K562 cells were released into fresh medium supplied with 10 mM HU, 50 μM BrdU, and 30 μM DRB for 12 h. Next, the cells were washed and released into fresh medium containing 10 mM HU for 3.5 h. For the G1 phase samples used in the DRB removal assay, G1-arrested K562 cells were cultured in the presence of 30 μM DRB for 12 h and released into fresh medium containing 5 μM palbociclib for an additional 3.5 h. Next, the fixed cells were denatured and incubated with antibodies against BrdU or γ-H2AX. For the RNase H1 overexpression assay, cells were transfected with plasmids containing GFP or GFP-RNase H1 before cell arrest was induced via palbociclib. Images were acquired using an LSM 710 NLO & DuoScan System (× 63, 1.4 NA or × 40, 1.2 NA) and analyzed using ImageJ according to the instructions included with the software.

## Quantification and statistical analysis
### Sequencing data analysis

**NAIL-seq data processing** R1 and R2 reads were stitched using *PEAR* [74] and the bridge adapter sequences were trimmed with custom scripts. The stitched reads were aligned to the human genome (assembly hg19) or mouse genome (assembly mm10) by *BWA-MEM* (-k 20). PCR duplicates were distinguished by random molecular barcodes (RMBs) and only one was kept. Reads with mapping quality (MAPQ) ≥ 30 were kept for further analysis. We used *samtools* to convert .sam files into .bam files.

**ChIP-seq data processing** First, we trimmed the adapter sequences in the reads using *cutadapt (-g NGGGGGGGGGG -g GGGGG --times=2 --minimum-length=70 --error-rate=0.2 -O 3)* [75]. Next, we aligned reads to the human genome (assembly hg19) using *BWA-MEM* (-k 20). For MCM ChIP-seq, we first generated RPKM to normalize the total reads by converting alignment files in .bam to .BigWig format using *bamCoverage* from *deeptools* [76] with parameters of *-bs 1000 and --normalizeUsing RPKM*. To compare with the control, the fold change of the RPKM signal between the MCM ChIP-seq and the input control was generated for further analysis. For ORC2 ChIP-seq, peaks were identified by *MACS 1.4.2* using -p 1e-5 --nolambda --nomodel --keep-dup=all [77]. The peaks within 500 bp were merged using BEDTools 2.27.0 [78]. Peaks within 500 bp from two biological replicates without α-amanitin treatment were defined as reproducible ORC2 peaks and used for further analysis. Genome-wide ORC2 tracks in .BigWig format were generated by bamCoverage from deeptools with -bs 50 and –normalizeUsing RPKM.

### Identifying early replication initiation zones (ERIZs) from NAIL-seq
We developed a pipeline named "RepFind" for NAIL-seq analysis, which is described in the following section.

**Defining E-B peaks from EdU/BrdU-labeled NAIL-seq samples** Both EdU and BrdU reads were normalized by reads per million (RPM) in 5-kb bins tiling the whole genome and marked as E and B, respectively. ΔEB was defined as: ΔEB=E-B. Bins, with ΔEB > 0.3, were taken as seeds and the neighbor bins with ΔEB > 0 were merged with the seeds to mark EdU-rich regions. The E-B peaks were the EdU-rich regions whose lengths were > 20 kb and fell in EdU peaks (defined from EdU reads using SICER with options -w 5000 -g 3). Peaks that occurred in at least two biological replicates were kept as bona fide E-B peaks.

**Defining EdU/HU peaks** The peak calling procedure was modified as reported previously [15]. Peaks were called by *MACS* (version 1.4.2) (*-p 1e-5 --nolambda --nomodel --keep-dup=all*). Next, we filtered the peaks with ≥ 400-fold enrichment against a random Poisson distribution generated from *MACS14*. Neighbor peaks were merged if their interval was shorter than 10 kb for K562 or 20 kb for GM12878. The merged peaks with < 10 kb size were discarded. Peaks in ChrY, mitochondria DNA, and black-list regions were omitted.

**Defining ERIZs** The ERIZs were defined as EdU/HU peaks that overlapped with E-B peaks, while non-ERIZs were defined as EdU/HU peaks that did not overlap with E-B peaks. The replication timing value was obtained from wavelet-smoothed signals from six fractions of the ENCODE Repli-seq profile. The mean value of replication timing for each region was calculated by *bwtool summary*. The distribution was plotted by ggplot2 histogram. Domains with a replication timing value > 0.5 were defined as early replication domains, and other domains were classified as late replication domains (Additional file 1: Figure S2c).

### Comparing OK-seq and SNS-seq with NAIL-seq

OK-seq data from K562 cells were analyzed using the BAMscale pipeline with default settings (https://github.com/ncbi/BAMscale/wiki/Detailed-Use:-OKseq-RFD-(Replication-Fork-Directionality)-Track-Generation). Replication initiation zones (IZs) were identified by using OKseq_switches.R with default settings (https://github.com/ncbi/BAMscale/wiki/Finding-OK-seq-strand-switched-from-the-RFD-track). Replication origins identified by SNS-seq were downloaded from GSE46189 (GSE46189_Ori-Peak.bed.gz). The overlaps between ERIZs/OK-seq and ERIZs/SNS-seq were calculated using *intersect* in Bedtools.

### Visualization of sequencing data

To generate genome tracks, we converted the .bam files into .BigWig files. For NAIL-seq and ORC2 ChIP-seq, files were generated by *bamCoverage* (in deepTools) with the parameter *--binSize 1000 --normalizeUsing RPKM*. For MCM5 ChIP-seq, the fold change value of the ChIP-ed samples over the input was generated by *bamCompare* with parameter *--binSize 1000, --normalizeUsing RPKM --operation ratio*. The data were presented by IGV 2.4.14 [79].

All heatmaps were generated by the R package "*EnrichedHeatmap*" [80]. For the non-TR heatmaps (see the "GRO-seq analysis" section for region definition), the 20–100-kb

non-TRs were centered at the midpoint and ordered by increasing width, and the ± 100-kb regions from the midpoint were used for display. Each region was binned with a 1-kb window filled with the mean value of each signal. For early replication or ORC2 ChIP-seq, the signal was generated by RPKM. Specially for MCM5 ChIP-seq, the z-score-transformed fold change value was displayed. For the transcribed gene-related heatmaps, ERIZ-flanked transcribed regions (defined by TR in the "GRO-seq" analysis) larger than 50 kb were ranked by gene width, with the smallest at the top. For display, all transcribed regions were scaled to the same width and aligned at both TSS and TTS. Additional 50-kb regions upstream or downstream of the TSS or TTS, respectively, were shown. The signal processing was performed in the same manner as that performed for the non-TR-related heatmaps.

### A/B compartment analysis

GM12878 and K562 HiC data (.hic file) were downloaded from GSE63525 [1]. Next, *eigenvector* from juicer [1] (KR BP 100000) was used to identify A/B compartments in GM12878 or K562 cells with 100-kb bins. The arbitrarily assigned "+" or "−" from eigenvector was corrected by defining "+" as an eigenvector positively correlated with an active histone marker (such as H3K27ac). The A compartments were defined as regions with a positive corrected value ("+") for the eigenvector.

### Prediction of ERIZ locations using epigenetic features

To predict ERIZ locations using epigenetic features within A compartments, we built a logistic regression model based on the peaks of the ENCODE ChIP-seq data (Fig. 2b; Additional file 1: Figure S3a). First, we converted the location of each ERIZ into a binary format using ChromHMM with 50-kb windows within an A compartment, where the window overlapping with the ERIZ is designated as "1" ("ERIZ-1"), and all others are designated as "0" ("ERIZ-0") [81]. Second, to generate the epigenetic predictor matrix for ERIZs, we defined the enrichment score of each 50-kb window using the log odds ratio of the observed and expected occurrence of ChIP-seq peaks, where the observed occurrence was defined as the proportion of 1-kb bins that overlapped with ChIP-seq peaks. The expected occurrence was estimated as the mean occurrence of the random peaks shuffled within the same chromosome. Third, we subsampled bins from the "ERIZ-0" bins to balance the sample size of the "ERIZ-1" bins. Finally, we built the logistic regression model using "glm" in R. The coefficients were used to evaluate the contribution to the prediction of ERIZ location.

### GRO-seq analysis

K562 and GM12878 GRO-seq data from a previous report were acquired for re-analysis [45]. For activated mouse B cells and mES V6.5 cells, GRO-seq data were acquired from published reports [15, 69]. Adapter-trimmed reads were aligned using *bowtie* (-l 25 -v 1 -k 1 -m 1 -S -q --best) [82]. Reads mapped to rRNA were omitted. For K562 and GM12878 cells, the read density (reads per kilobase, RPK) normalized to 10 million sequencing reads in the gene bodies or promoter regions was calculated via *analyzeRepeats.pl* from Homer (*analyzeRepeats.pl* rna hg19 -count genes -condense-Genes -strand + -norm 1e7) [83]. For activated mouse B cells and mES V6.5 cells, the

read density was obtained from Homer (*analyzeRepeats.pl* rna mm10 -count genes -condenseGenes -strand + -norm 1e6).

**Defining active genes** Active genes in K562 or GM12878 cells were defined as those with read density > 0 at the promoter region and RPK > 4 at the gene body from the GRO-seq data. Silent genes were defined as those with no read at the promoter region and RPK ≤ 1 at the gene body. Mouse active genes in activated B cells and mES V6.5 cells were defined as those with RPK > 1. Silent genes were genes with zero RPK from the GRO-seq data.

**Defining non-transcribed regions (non-TRs) and transcribed regions (TRs)** The non-transcribed regions used in heatmaps were defined as interval regions between two active genes within A compartments via *bedtools subtract*. Non-transcribed regions with a size of 20–100 kb that overlapped with ERIZs were used for the non-TR-related heatmaps. The active genes flanked by ERIZ-occupied non-TRs were defined as transcribed regions (TRs) for display in heatmaps (Fig. 4f–h).

**Comparing signal enrichment in non-transcribed regions and transcribed regions** The non-transcribed regions were further trimmed by 200 bp at the head and tail to exclude the TSS and TTS regions (Figs. 3b, d, f and 4c, e; Additional file 1: Figure S4f). The flanked transcribed regions were also trimmed as active gene body regions (TSS + 200 bp, TTS – 200 bp). The read density was calculated as follows: for each non-transcribed region, reads in the region were counted and then normalized by region length. Read counts in transcribed regions upstream or downstream of non-transcribed regions were summed and normalized by the summed length of the transcribed regions. In order to compare the signal enrichment of non-transcribed and transcribed regions based on NAIL-seq and ORC2 ChIP-seq, the log2 fold change of the ratio of the read density in the non-transcribed region to that of the transcribed region was calculated to allow quantification of differences. For MCM ChIP-seq, the fold enrichment of the read density relative to that of the input control was calculated and used to calculate the log2 fold change in the non-transcribed regions and transcribed regions.

### Analysis of dCas9 as a transcription barrier

To compare the read density adjacent to the dCas9 binding sites, the RPK ratio was defined as follows: the read count per kilobase (RPK) was calculated in each bin and then divided by the mean RPK of the B compartments at the same chromosome to normalize the background of each sample. For the *CMIP* locus, the ± 5-kb region centered on the dCas9 binding site was tiled in 1-kb bins (with a 200-bp sliding) for calculation of the RPK ratio (Fig. 5b, d). A 2-kb bin upstream or downstream of the dCas9 binding site was used to calculate the RPK ratios shown in Fig. 5c, e. For the *GALNT10* locus, the ± 6-kb region centered on the dCas9 binding sites was tiled in 3-kb bins (with a 300-bp sliding) (Additional file 1: Figure S6h).

## Statistical analysis

We performed Student's $t$ test to test the statistical significance between RPK ratios for dCas9 with gRNA-CMIP and dCas9 with scrambled gRNA. The $t$-test was applied to assess the difference between the dCas9-mediated transcription blockade and control groups in the MCM5 ChIP-qPCR assay. The $t$-test was also used to analyze the distribution of γ-H2AX intensity with or without DRB treatment. The Wilcoxon rank-sum test was performed to analyze the differences between the log fold change NAIL-seq signal and ORC2 or MCM5 ChIP-seq signals in non-transcribed regions in comparison with transcribed regions among the groups subjected to different treatments with transcription inhibitors.

## Supplementary Information

---

**Additional file 1: Figure S1.** Experimental procedure for NAIL-seq. **Figure S2.** Comparison among NAIL-seq, Repli-seq, OK-seq, SNS-seq, and EdU/HU-seq in K562 or mouse primary splenic B cells. **Figure S3.** The landscape of ERIZs. **Figure S4.** Transcription disturbance relocates ERIZs in K562 cells, mSBCs, and mESCs. **Figure S5.** Transcription inhibition does not lead to ORC2 relocation. **Figure S6.** dCas9-mediated transcription blockade induces MCM5 accumulation and replication initiation. **Figure S7.** Collision of transcription and DNA replication initiation induces genome instability. **Figure S8.** The full uncropped pictures for Western blots.

**Additional file 2: Table S1.** The locations of ERIZs in GM12878 cells, aligned to hg19.

**Additional file 3: Table S2.** The locations of ERIZs in K562 cells, aligned to hg19.

**Additional file 4: Table S3.** The locations of ERIZs in mESCs, aligned to mm10.

**Additional file 5: Table S4.** Key resource table.

**Additional file 6.** Review history.

---

### Acknowledgements

We acknowledge the members of the Hu laboratory for their helpful discussions. The GFP-RNaseH1 plasmid was a kind gift from Dr. Sijie Liu and Dr. Daochun Kong. We thank the Flow Cytometry Core at the National Center for Protein Sciences, Peking University, particularly Liying Du, for her technical help. We also thank Dr. Siying Qin at the Core Facilities of the School of Life Sciences, Peking University for assistance with optical/confocal imaging.

### Review history

The review history is available as Additional file 6.

### Peer review information

### Authors' contributions

Y.L., C.A., and J.H. developed NAIL-seq; C.A. wrote the "RepFind" pipeline for NAIL-seq analysis; Y.L., C.A., T.G., and J.H. designed the experiments; Y.L., T.G., R.L., J.W. Y.J., and X.L. performed the experiments; Y.L., C.A., T.G., N.G., Q.L., X.J., and J.H. analyzed the data; Y.L., C.A., T.G., N.G., Q.L., X.J., and J.H. wrote the paper. The author(s) read and approved the final manuscript.

### Funding

This work was supported by the National Key R&D Program of China (2017YFA0506700 to J.H., 2017YFA0506600 to X.J., 2019YFA0508900 and 2016YFA0500700 to N.G.) and the NSFC grant (31771485 to J.H., 31871309 to X.J., 31830048 to Q.L., 31725007, and 31630087 to N.G.). N.G., Q.L., X.J., and J.H. are investigators at the PKU-TSU Center for Life Sciences. J.H. is also a Bayer Investigator and Y.L. is supported by the Boehringer Ingelheim-Peking University Postdoctoral Program.

### Availability of data and materials

All sequencing data presented in the present study were deposited at NODE (National Omics Data Encyclopedia) under the accession number of OEP000658 [84] and were also deposited in NCBI Gene Expression Omnibus (GEO) database under accession GSE174680 [85]. All the other data generated in the present study can be found in the manuscript and additional files. All public data used in this study have been listed and well referenced in Additional file 5: Table S4, including Repli-seq of K562: GSM923448 [19], SNS-seq of K562: GSE46189 [23], ORC2 ChIP-seq from K562: GSE70165 [31], Repli-seq of GM12878: GSM923451 [86], MCM3 ChIP-seq from K562: GSE92216 [86], MCM5 ChIP-seq from K562: GSE127403 [86], HiC from K562 and GM12878: GSE63525 [1], GRO-seq for GM12878: GSM1480326 [45], GRO-seq for K562: GSM1480325 [45], G4 ChIP-seq from K562: GSM2876090 [87], EdU-seq-HU in mouse splenic B cells: GSE116321 [15], nsRNA-seq for mouse splenic B cells: GSM3227964 [15], and GRO-seq for mESC: GSM1665566 [69] in GEO database and OK-seq of K562: PRJEB25180 [24] in European Nucleotide Archive database. Scripts of RepFind

pipeline for NAIL-seq in this study are available in GitHub repository under GNU general public license Version 3 [88] and in Zenodo [89]. All cell lines used in this study have been authenticated and are available upon request.

## Declarations

### Ethics approval and consent to participate
Not applicable.

### Consent for publication
Not applicable.

### Competing interests
The authors declare no competing interests.

### Author details
[1]The MOE Key Laboratory of Cell Proliferation and Differentiation, School of Life Sciences, Genome Editing Research Center, Peking University, Beijing 100871, China. [2]Peking-Tsinghua Center for Life Sciences, Peking University, Beijing 100871, China. [3]State Key Laboratory of Membrane Biology, School of Life Sciences, Peking University, Beijing 100871, China. [4]State Key Laboratory of Protein and Plant Gene Research, School of Life Sciences, Peking University, Beijing 100871, China.

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

## 
