## [**Additional file 6.** Review history. · Genome Biology]

Review History

First round of review

Reviewer 1

Are you able to assess all statistics in the manuscript, including the appropriateness of statistical tests used? No, I do not feel adequately qualified to assess the statistics.

Comments to author:

Liu et al present a genome-wide survey of early DNA replication in cultured mammalian cells. The study is timely. In the last two decades, the molecular fundamentals of eukaryotic DNA replication have been described and the mammalian genome replication program characterised. Mammalian origins have been mapped, although this issue remains challenging. Lui et al. make an attempt at intensifying these characterisations with higher resolution and by addressing the mechanistic basis based on earlier suggested models how transcription shapes the replication landscape.

First, the authors develop a higher resolution method (NAIL) for detecting early replication (ERIZ) in cultured mammalian cells by using synchronous S phase entry and consecutive labelling by two separate nucleotide analogues. Then, the authors characterise early replication in these cells, reporting ERIZ at chromatin loop bases and outside actively transcribed genes, despite general correlation of ERIZ with transcription activity and the chromatin A compartment. The authors then investigate the relation of transcription and replication deeper by various approaches using manipulation of transcription both genome-wide and gene-specific to study the consequences on ERIZ distribution. They suggest that transcription elongation re-locates pre-RCs (potential replication origins) from gene bodies into gene-adjacent regions.

The study interrogates previously proposed but (in higher eukaryotes) little tested mechanisms rather than presenting new models that shape the replication landscape. The developed NAIL technique (Fig 1) looks to yield indeed higher resolution ERIZ mapping than previous mapping (although this is difficult to judge for a non-expert in such techniques based on the presented data). The presented relation between ERIZ, transcription and chromatin domains (Figs 2,3) confirm previous findings but present little new insight if any. Figs 4-6 then investigate transcription-based mechanisms to shape the replication landscape by A) using pan-transcription inhibitors as well as bypassing of termination, B) Genetically inactivating promoters of specific genes, and C) putting transcription roadblocks into specific genes using nucleolytically inactive Cas9 targeted to specific genes. These approaches are adequate to gain mechanistic insight how the replication landscape is established and do provide interesting new aspects. However, this set of data seems in parts immature. Last, they present a single experiment suggesting that the observed replication characteristics are required to prevent replication-dependent DNA damage.

I suggest to shorten and focus the paper and to the experiments of the remaining part. The presented research must be clearer described and, importantly, better discussed in light of the present literature. The model presented in Fig 7 is not fully supported by experimental evidence and needs revision.

Major points:

1) Description of presented experiments unclear.

It is difficult to completely understand the experiments presented. The authors should make sure that all experiments can be followed by reading the main text and figure legends. Often it is unclear how

experiments were exactly done. Moreover, the figure legends often do not present a clear description of how the data was generated. The legends typically do not name the method used nor describe how cells were exactly treated.

2) Focussing and shortening of the manuscript

- Figures 2 and 3 present little novel insight: the correlation of early replication with compartment A and transcription, the association of ERIZ with CTCF binding sites at the base of loops and the anti-correlation with transcribed gene bodies (despite overall correlation with highly transcribed compartment A) were described, for example by the Hyrien lab. Some or most of these experiments are still useful for the current manuscript in order to characterise the method of ERIZ used in the paper. The data should mostly go in the supplementary information section and be presented short. If the authors think they do present novel or more detailed aspects they should focus on these aspects and more clearly distinguish in the text what exactly is new.

- Data presented in Fig 4 is not convincing.

I appreciate the approach to manipulate transcription initiation and termination to learn about how transcription influences replication. However, the effects presented in 4d-e and the corresponding supplementary figures seem minute. The effect of inhibiting transcription of a gene in 4b is more convincing. However, it remains unaddressed whether the ERIZ signal upon in GALNT10^{-/-} cells reflects a change of replication initiation or replication elongation (caused, for example, by collision between replication and transcription machines) in response to switching off transcription. Only changes in initiation would fit to the proposed models.

Thus, showing these data would require significantly better characterisation of the systems used. I therefore suggest to remove these figures from the manuscript and improve the part presented in figures 1 and 5-7.

3) Extending and strengthening experiments of figures 1 and 5-7.

- Figure 1 introduces NAIL to identify ERIZ. The experiments seem thorough and the data presented convincing, but would benefit from some control experiments.

Key is to show that ERIZ represents exclusively early firing origins. The authors use HU to identify replication from dormant origin upon fork stalling. High resolution is gained from immunoprecipitating two nucleoside analogues (BrdU and EdU) and subtracting the BrdU signals from the EdU signals (E-B). ERIZ is defined as peaks in E-B and EdU/HU samples (but not EdU/HU-only).

Puzzling to me was to realise that in many subsequent experiments the authors seem to use EdU/HU data, not E-B. As stated above, please make clear in the text or legend which method was used. If my impression is true please deal with this more openly: discuss resolution by both methods (EdU/HU vs E-B) and state a (presumably) lower resolution (EdU/HU) is acceptable in most experiments.

Connected the the point made about resolution of the mapping: What's the resolution in E-B and EdU/HU really compared to more classic methods?

As stated above, showing that exclusively early origins are called is key. More clearly state the evidence presented that only early origins are identified. To strengthen this point, absence of signals in late replicating regions should be shown: a quantitative statement how many known early AND how many late regions are picked up would do the job. In addition, caffeine or other checkpoint inhibitors (+HU) could help define early origins as such origins that are not inhibitable by the checkpoint when cells enter S phase (the classic definition in budding yeast).

A better scale should be used in 1b to better judge the resolution of the data. Also, it is not described in the legend or main text which time point of G1-release was used for this experiment.

- Figure 5 supports the model that transcription redistributes ERIZ through re-locating pre-RCs (but not ORC). The figure suggests that global inhibition of transcription re-distributes ERIZ and Mcm5, but not ORC.

5b shows a dramatic distribution change of ERIZ in response to alpha-amanitin treatment. Treatment with DRB was done in an attempt to address potential unspecific effects of the amanitin. DRB shows a similar, albeit a milder, effect on ERIZ. The authors please make a statement about how this milder DRB effect relates to DRB effectiveness of transcription inhibition compared to amanitin.

A conceptually important point that is unaddressed is to which locations ERIZ re-distributes in response to transcription inhibition. If re-distribution by transcription machines is involved it is to be expected that ERIZ is in transcribed gene bodies upon transcription inhibition. In 5b, genome regions were centred on the sites between adjacent genes, but where gene bodies are is not indicated. Do ERIZ signals reside in gene bodies after amanitin treatment?

Redistribution of fired origins away from ORC sites is another key prediction of the model, independent of whether the re-distribution occurs through transcription or not. The authors address this in 5d. ERIZ (5b) seem to re-distribute upon amanitin to sites where ORC resides (5d). Can this be concluded from the data? Please state.

5f suggests that amanitin re-distributes Mcm5, consistent with the model that transcription re-distributes ERIZ via re-distributing pre-RCs (but not ORC). Presumably, Mcm5, which is measured in 5f, is a proxy for pre-RCs. Validity of this assumption could be tested using siRNA against Cdc6 or Cdt1 (or other means to inhibit origin licensing).

Conceptually, it is important to understand where pre-RCs re-locate in response to amanitin. The assumption must be that it is situated 1) more in gene bodies, and 2) close to ERIZ signals (5b). These effects cannot be seen in 5f. I presume that the fact that pre-RCs are probably present in such high amounts on chromatin blurs Mcm5 signals, making a statement about specific localisation difficult. In light of this, it is sufficient to show that a specific localisation between genes is lost upon amanitin. Can the author please discuss this important point clearer?

- Figure 6 uses a specific block of the transcription machinery to provoke origin firing close to the block, which is predicted if transcription re-located pre-RCs. More specifically, the prediction must be that origins fire downstream of the block if the model is that transcription complexes push pre-RCs from the gene body.

6c shows a strong effect of the targeted Cas9, suggesting a change of ERIZ location. What remains unclear to me is if ERIZ is now found in the gene body. A better labelling would help. It seems to me that ERIZ is upstream of the block, not downstream as would be expected on the basis of a pushing model. Can the authors discuss this in the main text please? Confirmation experiments in Fig S8 (for example f) using an independent locus do not seem to have been done at the necessary resolution/quality and should be improved.

- Figure 7a and b) is used to suggest that transcription-replication collision in gene bodies in DRB-treated cells leads to DNA damage. To make this statement, the authors must 1) confirm the effect by using amanitin, and 2) address if the DNA damage detected depends on replication-transcription collision. The damage is clearly S phase -specific, but it could result from other mechanisms than collision, such as lack of production of histones or other replication factors. RNase H could be used to address dependency on transcription assuming that R-loops play a role in replication-dependent damage generation upon transcription inhibition. Another approach would be to test if DRB/amanitin produce less damage in late S phase where predominantly non-transcribed regions are replicated.

- Model in Fig 7c. Although intuitive from the existing published models the presented model is not fully supported by the data shown. Re-distribution of ERIZ (and pre-RCs) is shown, but not pushing of pre-RCs by transcription machines. Alternative explanations like dissociation of pre-RCs are not discussed. Sliding of pre-RCs on naked DNA was shown by the Diffley, Remus and Speck labs, but whether it occurs on chromatin less clear.

- Discussion:

The discussion is very short and should more comprehensively discuss the presented experiments in light of the existing literature.

Reviewer 2

Are you able to assess all statistics in the manuscript, including the appropriateness of statistical tests used? No, I do not feel adequately qualified to assess the statistics.

Comments to author:

In this manuscript, Liu et al. explore the link between transcription and replication origin activity in human cells -- specifically, the long-standing observation that origin activity in early S phase is reduced or absent in actively transcribed regions. The first part of the manuscript focuses on establishing their genome-wide replication assay, pulse-labeling with nucleoside analogues followed by sequencing of the newly replicated DNA, in two human cell lines. They find that early origin activity is excluded in actively transcribed genes but not in transcriptionally silent genes, a correlation that is supported by epigenetic marks of active vs. silent chromatin. Of particular interest is that H2A.Z very strongly correlates with early initiation sites, again consistent with the recent literature on the role of H2A.Z in recruiting ORC1.

This portion of the manuscript mostly recapitulates the previously documented negative correlation between transcription and early origin firing, suggestive of a mechanism where active transcription prevents origin firing. What is newer is that the authors then go on to perturb transcription *in vivo* to test the specific hypothesis that active transcription pushes assembled MCM complexes out of the transcribed regions so that they end up in non-transcribed regions. Initiation of DNA synthesis, which occurs at the MCM complexes, would therefore be excluded from transcribed regions. They use three approaches to test this hypothesis -- inhibition of polII-driven transcription with alpha-amanitin, read-through transcription using a nuclease-deficient allele of XRN2, and blocking transcription using dCas9. In all three, they examine origin activity as well as locations of ORC and MCM. Their model predicts that limiting transcription should expand the zones of early origin firing, whereas read-through transcription should further restrict early replication initiation sites.

For the most part, their results are consistent with their model. However, there were several areas where they did not provide sufficient detail to adequately evaluate the data (e.g., the pairs of tandem genes they looked at in their read-through transcription experiment), and there were controls that should be included that either weren't done or weren't shown (e.g., validating the transcription block; see specific comments below). In fact, their manuscript overall needs more detail on logic and expectations -- e.g., stating more clearly what outcomes they expect to see (with regard to the figures they show) if their hypothesis is correct vs. if it is not. If suitably revised and with the additional controls, this work can be a nice addition to existing literature on transcription vs. origin activation.

Specific comments (in no particular order)

1. The authors make much of the EdU/BrdU-seq method, but it is not at all clear what the benefit of this method is over the more conventional EdU-seq. The best resolution they get is either by doing EdU/HU-seq (as has been done by others in the past) or by subtracting the BrdU signal from the EdU+BrdU signal... in other words, they seem to be adding BrdU signal and then subtracting it out,

which just seems to be introducing unnecessary complexity and data processing. Why not just do a short EdU pulse? More explanation/justification is needed.

2. On a related note, they show comparisons of their data with repli-seq and Okazaki-seq from the literature. However, they don't show comparisons with EdU/HU-seq (e.g., from the Macheret and Halazonetis paper, ref. 14), which would seem the closest comparison. They should include such comparisons.

3. In their EdU/HU-seq plots (e.g., Fig. 1b), the peaks appear to be split, as though the signal is representing forks that have already diverged (ie., forks at the two ends of the replication bubble). Is that the appropriate interpretation? If so, are they not capturing initiation events early enough? More explanation is needed. Also, at the bottom of p.5, the authors note that multiple SNS-seq peaks fell into a single early replication initiation zone identified in this manuscript (Fig. S2a). They should mark those locations clearly in the figure, or give the coordinates they are talking about in the text, so that the reader can see which peaks they are describing.

4. A minor point, they refer to "peak length" in figures 1c and S2b. Perhaps "peak width" would be a more conventional term.

5. In some figures, graphs that we should be able to compare directly are plotted on different scales, which can be misleading (e.g., Fig. 3b and c, Fig 5f). Plots should be on the same scale. For example, if control and amanitin-treated samples in Fig. 5f are plotted on the same scale, the apparent difference might disappear or not be as pronounced. In fact, looking at the summary plot in Fig. 5g, it does appear that the difference, although significant, is slight.

6. In figures 4 and 5 (looking at 410 pairs of co-directionally transcribed genes), it is not clear what the distribution of spacing in the gene pairs is. They should provide a histogram of the inter-gene distances, as this distribution affects the interpretation of the data. For example, in the polII ChIP-seq profiles in Fig 4e, what leads to the very distinct hills-and-valleys shape downstream of the TTS? In how many of the 410 pairs is the second gene beyond the 150 kb region that is plotted, and in how many is the second gene (transcribed region) within the 150 kb downstream of the TTS that is shown?

7. Since they must already have the data, they should also show, for comparison, the polII and early replication data for convergently transcribed genes (where there should be a shift) and in oppositely-oriented genes (aligned by TSS) where according to their model there should not be a shift. Given the small shifts seen in their data, having these additional controls would be helpful. In fact, in Fig. S5g it appears that read-through transcription shifts the bulk of the EdU replication signal towards the gene and not away from the gene as would be expected of their model.

8. Fig. 4a and 4c are not very helpful and can be eliminated.

9. Fig. 5b -- why is the EdU replication signal more tightly resolved in the amanitin-treated samples? Is this a consequence of slower forks, or a change in S phase progression? Or is it because the control (untreated) plot is not to the same color scale as the experimental sample plots? More explanation and setup (clear statement of expectations) is needed. Also, for Fig. 5f and g, they should say more clearly what "fold change" means.

10. The authors should consider showing a plot of the difference between the ORC signal and MCM signal. If ORC is unperturbed but MCM does shift, we should see a greater separation between the MCM and ORC signal locations in the perturbed vs. control (unperturbed).

11. For the transcription blockage using dCas9, just showing the overall level of transcription does not adequately validate that the block is working as expected. They need to show RT-qPCR signal (or polII occupancy) to the left vs. the right of the block to show that there is no change upstream of the block vs. reduced transcription downstream of the block. "Relative transcription level" (Fig. 6b) is

cryptic, it is not clear relative to what. It is unclear how the quantification for Fig. 6g was done, more detail is needed. Also, in Fig. 6d and f, the symbol for gRNA-CMIP is missing in the legends.

12. The authors should consider moving the microscopy images in Fig. S9 to the main figure (Fig. 7).

13. In the Introduction (p.4, lines 86-88) the authors state that "...suppression of RNA polymerase I, the primary ribosomal DNA transcription polymerase, also causes MCM relocation at the rDNA locus in budding yeast [38]." That statement is not correct, what that paper showed was that suppression of silencing of polII-driven transcription in the rDNA spacer led to MCM relocation.

14. On p.6, line 145: "precious" should be "previous".

15. On the same page, line 156, and Fig. 2: it is unclear what the authors mean by "intra-compartment activity". Are they talking about the level of transcription within the compartment? Or the frequency of contacts within the compartment in 3D space? "The levels of ERIZ-associated early replication ranked from low to high in the same order as the compartment activity and transcription level" (lines 157-159) implies that "compartment activity" is a separate phenomenon than transcription. But, "These data indicate that transcription in the active A compartments is coincident with early DNA replication initiation" (lines 161-162) suggests that "intra-compartment activity" refers to transcription activity. Again, more explanation is needed.

Reviewer #1: review:

Liu et al present a genome-wide survey of early DNA replication in cultured mammalian cells. The study is timely. In the last two decades, the molecular fundamentals of eukaryotic DNA replication have been described and the mammalian genome replication program characterised. Mammalian origins have been mapped, although this issue remains challenging. Lui et al. make an attempt at intensifying these characterisations with higher resolution and by addressing the mechanistic basis based on earlier suggested models how transcription shapes the replication landscape.

First, the authors develop a higher resolution method (NAIL) for detecting early replication (ERIZ) in cultured mammalian cells by using synchronous S phase entry and consecutive labelling by two separate nucleotide analogues. Then, the authors characterise early replication in these cells, reporting ERIZ at chromatin loop bases and outside actively transcribed genes, despite general correlation of ERIZ with transcription activity and the chromatin A compartment. The authors then investigate the relation of transcription and replication deeper by various approaches using manipulation of transcription both genome-wide and gene-specific to study the consequences on ERIZ distribution. They suggest that transcription elongation re-locates pre-RCs (potential replication origins) from gene bodies into gene-adjacent regions.

The study interrogates previously proposed but (in higher eukaryotes) little tested mechanisms rather than presenting new models that shape the replication landscape. The developed NAIL technique (Fig 1) looks to yield indeed higher resolution ERIZ mapping than previous mapping (although this is difficult to judge for a non-expert in such techniques based on the presented data). The presented relation between ERIZ, transcription and chromatin domains (Figs 2,3) confirm previous findings but present little new insight if any. Figs 4-6 then investigate transcription-based mechanisms to shape the replication landscape by A) using pan-transcription inhibitors as well as bypassing of termination, B) Genetically inactivating promoters of specific genes, and C) putting transcription roadblocks into specific genes using nucleolytically inactive Cas9 targeted to specific genes. These approaches are adequate to gain mechanistic insight how the replication landscape is established and do provide interesting new aspects. However, this set of data seems in parts immature. Last, they present a single experiment suggesting that the observed replication characteristics are required to prevent replication-dependent DNA damage.

I suggest to shorten and focus the paper and to the experiments of the remaining part. The presented research must be clearer described and, importantly, better discussed in light of the present literature. The model presented in Fig 7 is not fully supported by experimental evidence and needs revision.

We appreciate the reviewer's constructive comments, which are very helpful in improving our manuscript. We revised the manuscript following the reviewer's suggestions and hope that the reviewer finds the revised manuscript acceptable for publication in Genome Biology. Briefly, we reorganized the manuscript and make it more clearly focused on the mechanism of transcription-mediated the redistribution of early replication initiation. We provided more evidence to support our findings and models, including:

- 1) We employed the degron system to validate the infiltration of DNA replication initiation into transcribed regions in the absence of RNA polymerase II in mouse embryonic cells;

- 2) We found the amounts of MCM is constant in the chromatin and redistributed to the transcribed regions under transcription inhibition;

3) We performed the RNase H1 over-expression experiments to confirm the collisions of early replication with transcription as suggested.

More details are described below.

Major points:

1) Description of presented experiments unclear.

It is difficult to completely understand the experiments presented. The authors should make sure that all experiments can be followed by reading the main text and figure legends. Often it is unclear how experiments were exactly done. Moreover, the figure legends often do not present a clear description of how the data was generated. The legends typically do not name the method used nor describe how cells were exactly treated.

We apologize for the inconvenience. In the revised version, we moved the experimental details from the “Material and methods” section to the main text and figure legends. We also added more details including the experimental procedures and the details for analysis to make the figure easy to follow.

2) Focussing and shortening of the manuscript

- Figures 2 and 3 present little novel insight: the correlation of early replication with compartment A and transcription, the association of ERIZ with CTCF binding sites at the base of loops and the anti-correlation with transcribed gene bodies (despite overall correlation with highly transcribed compartment A) were described, for example by the Hyrien lab. Some or most of these experiments are still useful for the current manuscript in order to characterise the method of ERIZ used in the paper. The data should mostly go in the supplementary information section and be presented short. If the authors think they do present novel or more detailed aspects they should focus on these aspects and more clearly distinguish in the text what exactly is new.

Thanks for the helpful comments. In the revised manuscript, we have condensed and merged Fig. 2&3 to make a new figure focusing on the relationship between early replication initiation and transcription in human (a cancer cell line, K562, and primary-like cell line, GM12878) and mouse (mouse embryonic cell, mESC, and primary splenic B cells, freshly isolated from mouse spleen) cells.

- Data presented in Fig 4 is not convincing.

I appreciate the approach to manipulate transcription initiation and termination to learn about how transcription influences replication. However, the effects presented in 4d-e and the corresponding supplementary figures seem minute. The effect of inhibiting transcription of a gene in 4b is more convincing. However, it remains unaddressed whether the ERIZ signal upon in GALNT10^{-/-} cells reflects a change of replication initiation or replication elongation (caused, for example, by collision between replication and transcription machines) in response to switching off transcription. Only changes in initiation would fit to the proposed models.

Thus, showing these data would require significantly better characterisation of the systems used. I therefore suggest to remove these figures from the manuscript and improve the part presented in figures 1 and 5-7.

We appreciate the reviewer’s helpful comments and removed these data in the revised manuscript. We agree with the reviewer that the phenotype presented in the original Fig 4 is mild, and these data by themselves are not sufficient to prove the model and need to be combined with other data. Instead, we employed the degron system to validate the infiltration

of DNA replication initiation into transcribed regions in the absence of RNA polymerase II in mouse embryonic cells, validating our findings with transcription inhibitors in K562 cells (Fig. 3).

3) Extending and strengthening experiments of figures 1 and 5-7.

We added more analysis and also provided new evidence in the revised manuscript, including:

1) employing the degron system to validate the infiltration of DNA replication initiation into transcribed regions in the absence of RNA polymerase II in mouse embryonic cells;

2) We found the amounts of MCM is constant in the chromatin and redistributed to the transcribed regions under transcription inhibition;

3) We performed the RNase H1 over-expression experiments to confirm the collisions of early replication with transcription as suggested.

See below for more details.

- Figure 1 introduces NAIL to identify ERIZ. The experiments seem thorough and the data presented convincing, but would benefit from some control experiments.

Key is to show that ERIZ represents exclusively early firing origins. The authors use HU to identify replication from dormant origin upon fork stalling. High resolution is gained from immunoprecipitating two nucleoside analogues (BrdU and EdU) and subtracting the BrdU signals from the EdU signals (E-B). ERIZ is defined as peaks in E-B and EdU/HU samples (but not EdU/HU-only).

Puzzling to me was to realise that in many subsequent experiments the authors seem to use EdU/HU data, not E-B. As stated above, please make clear in the text or legend which method was used. If my impression is true please deal with this more openly: discuss resolution by both methods (EdU/HU vs E-B) and state a (presumably) lower resolution (EdU/HU) is acceptable in most experiments.

The resolution from high to low in order is: EdU/HU > E-B > E or B > Repli-seq (Fig. 1b, 1c, and S2a, etc.). We described these comparisons in the Results and Discussion sections in the revised manuscript as the reviewer suggested. Since EdU/HU has a higher resolution and the signals are natural (with minimal processing), we extracted EdU/HU signals from the ERIZ-associated regions in the subsequent experiments for analysis. We indicated the resources of the presented data in the revised manuscript as suggested.

Connected the the point made about resolution of the mapping: What's the resolution in E-B and EdU/HU really compared to more classic methods?

As shown in Fig. 1c and Fig. S2b, the narrowest peaks from E-B and EdU/HU are less than 10 kb, while the narrowest peak from Repli-seq is over 100 kb. Moreover, the median peak width from E-B, EdU/HU, or Repli-seq is 90, 55, or 340 kb, respectively. We have added these sentences in the Results section of the revised manuscript.

As stated above, showing that exclusively early origins are called is key. More clearly state the evidence presented that only early origins are identified. To strengthen this point, absence of signals in late replicating regions should be shown: a quantitative statement how many known early AND how many late regions are picked up would do the job. In addition, caffeine or other checkpoint inhibitors (+HU) could help define early origins as such origins that are not inhibitable by the checkpoint when cells enter S phase (the classic definition in budding yeast).

Thanks, and sorry for the confusion. Hydroxyurea (HU) has the potential in inducing DNA damages or early firing of late origins, but the E-B signals are from cells under G1/S

transition without replication stress. Therefore, there should not contain replication signals from late origins in the E-B samples. The overlapped analysis help remove late origins from EdU/HU samples by E-B samples (Fig. 1e), and we found that 99.3% or 95.5% of identified ERIZs fall in the early replication domains of K562 or GM12878, respectively (Fig. S2c in the revised manuscript). We also revised the sentence as “HU treatment yields high-resolution replication-associated peaks but may induce extra DNA double-stranded breaks (DSBs) and early utilization of late DNA replication origins [39-42]; while the E-B signals are from cells under G1/S transition without replication stress, therefore E-B signals would help discriminate early firing of late origins and potential DNA damages in the EdU/HU libraries.” to make this point clear.

A better scale should be used in 1b to better judge the resolution of the data. Also, it is not described in the legend or main text which time point of G1-release was used for this experiment.

We have shown a larger scale in the revised Fig. 1b and the E-B data are yield at 2.5 hour after G1-release when the K562 cells are entering G1/S transition. The data of EdU/HU is obtained after release from G1 for 12 hours. We have included the description in the revised figure legends and main text.

- Figure 5 supports the model that transcription redistributes ERIZ through re-locating pre-RCs (but not ORC). The figure suggests that global inhibition of transcription re-distributes ERIZ and Mcm5, but not ORC.

5b shows a dramatic distribution change of ERIZ in response to alpha-amanitin treatment. Treatment with DRB was done in an attempt to address potential unspecific effects of the amanitin. DRB shows a similar, albeit a milder, effect on ERIZ. The authors please make a statement about how this milder DRB effect relates to DRB effectiveness of transcription inhibition compared to amanitin.

The two inhibitors have different mechanisms for transcription inhibition. α -amanitin binds to RNA polymerase II directly to stall transcription, while DRB suppresses the activation of RNA polymerase II by blocking CDK9 phosphorylation on RNA polymerase II. Moreover, the inhibition of α -amanitin is irreversible while the inhibition of DRB is reversible (Bensaude, 2011). For these reasons, we used the two inhibitors for transcription perturbation and detected similar but varied levels of changes of early replication signals. And for the same reason, we only used DRB but not α -amanitin in Fig. 6 (original Fig. 7) to induce transcription restart for detecting DNA damages. To strengthen our conclusion about the redistribution of early replication initiation by transcription, we used the IAA-induced degron system to degrade RNA polymerase II in a short time, 4 hours, and detected a significant decrease of early replication signals in the non-transcribed regions of mESCs (Fig. 3e-g).

A conceptually important point that is unaddressed is to which locations ERIZ re-distributes in response to transcription inhibition. If re-distribution by transcription machines is involved it is to be expected that ERIZ is in transcribed gene bodies upon transcription inhibition. In 5b, genome regions were centred on the sites between adjacent genes, but where gene bodies are is not indicated. Do ERIZ signals reside in gene bodies after amanitin treatment?

To better present the changes of early replication, ORC2, and MCM5 signals, we aligned the ERIZ-adjacent transcribed genes at both transcription start sites (TSS) and transcription termination sites (TTS) to perform heatmap analysis. As showed in Fig. 4f-h of the revised manuscript, we detected increased early replication signals and MCM5 within transcribed genes following transcription inhibition.

Redistribution of fired origins away from ORC sites is another key prediction of the model, independent of whether the re-distribution occurs through transcription or not. The authors address this in 5d. ERIZ (5b) seem to re-distribute upon amanitin to sites where ORC resides (5d). Can this be concluded from the data? Please state.

As shown in Fig. 4f and 4g of the revised manuscript and the figure below, we found that early replication initiation is relocated to ORC binding sites upon α -amanitin treatment. We have added the sentence “In particular, the α -amanitin treatment resulted in similar distribution patterns of ERIZs compared with ORC2” in the revised main text.

5f suggests that amanitin re-distributes Mcm5, consistent with the model that transcription re-distributes ERIZ via re-distributing pre-RCs (but not ORC). Presumably, Mcm5, which is measured in 5f, is a proxy for pre-RCs. Validity of this assumption could be tested using siRNA against Cdc6 or Cdt1 (or other means to inhibit origin licensing).

We hope we understand this comment correctly -- Elimination of CDC6 or CDT1 blocks the reloading of MCM onto chromatin. In this context, we should detect a decrease of chromatin-bound MCM if transcription is only responsible for dissociating MCM from the transcribed regions. Therefore, to test this possibility, we employed shRNA to knockdown CDC6 or CDT1 in G1-arrested K562 cells as suggested. We arrested the K562 cells in the G1 phase, taking 36 hours, and then transfected with the shRNA-containing plasmids for additional incubation of 24 hours. We checked the amounts of pre-RC components in whole-cell extracts (WCE) or chromatin extracts (Chro) and found the chromatin-bound ORC and MCM are constant after CDT1 knock-down (see below). However, the knock-down efficiencies of CDT1 in WCE only reach <35% and the amounts of Chro CDT1 are relatively constant with or without shRNA transfection. Moreover, the two shRNA cannot reduce the amounts of CDC6 in G1-arrested K562 cells (right panel). Therefore, we cannot draw a convincing conclusion from these data to exclude the above possibility. We emphasized this alternative possibility (transcription dissociates MCM) in the Discussion section of the revised manuscript.

shRNA	Sequence (5'-3')	References
shCdt1	AGGATGCTGGGGAGTCCTGCA	Zhang et al., PNAS, 2011
shCdt1	CGGAGCGTCTTTGTGTCGGAA	Varma et al., Nat Cell Biol, 2012
shCdc6	AGGCACTTGCTACCAGCAAGC	MISSION® shRNA Plasmid DNA (#SHCLND-NM_001254), Sigma
shCdc6	AGCTATTGCTCAGGAGATTTG	

Conceptually, it is important to understand where pre-RCs re-locate in response to amanitin. The assumption must be that it is situated 1) more in gene bodies, and 2) close to ERIZ signals (5b). These effects cannot be seen in 5f. I presume that the fact that pre-RCs are probably present in such high amounts on chromatin blurs Mcm5 signals, making a statement about specific localisation difficult. In light of this, it is sufficient to show that a specific localisation between genes is lost upon amanitin. Can the author please discuss this important point clearer?

Thanks for the insightful comment. It's difficult to detect the minor change of MCM in specific locus due to the widely-distributed high amount of MCM (both active or inactive), and the low quality of existing antibodies against MCM. However, the fold-change analysis, the same definition as *ENCODE* project and a previous report (Powell et al., 2015) (see Figure legends or Methods for details), showed us some clues. In the revised manuscript, we aligned the ERIZ-adjacent transcribed genes at both TSS and TTS. We found an increase of MCM5 within transcribed genes following transcription inhibition (Fig. 4e and 4h).

- Figure 6 uses a specific block of the transcription machinery to provoke origin firing close to the block, which is predicted if transcription re-located pre-RCs. More specifically, the prediction must be that origins fire downstream of the block if the model is that transcription complexes push pre-RCs from the gene body.

Thanks for the comment. From our model, RNA polymerase II-associated MCM accumulates right upstream of the dCas9 binding sites and induces new replication initiation as shown in Fig.5 and Fig. S6. While for the loaded MCM if any in downstream, MCM would accumulate at the loading sites (ORC binding sites) to initiate DNA replication initiation. There is no obvious ORC binding site downstream of the dCas9 binding sites on the *CMIP* gene (Fig. S6c). There are two presumable ORC binding hotspots downstream of the dCas9 binding sites on *GALNT10*, and a low level of replication signals occurs around these sites with dCas9 blockade, especially in repeats 2 and 3 (Fig S6f and S6g).

6c shows a strong effect of the targeted Cas9, suggesting a change of ERIZ location. What remains unclear to me is if ERIZ is now found in the gene body. A better labelling would help. It seems to me that ERIZ is upstream of the block, not downstream as would be expected on the basis of a pushing model. Can the authors discuss this in the main text please? Confirmation experiments in Fig S8 (for example f) using an independent locus do not seem to have been done at the necessary resolution/quality and should be improved.

Thanks for the comment. From our model, if RNA polymerase II pushes MCM along the transcribed genes, the stalling of RNA polymerase II by dCas9 would also induce the stalling and accumulation of MCM right upstream of the dCas9 binding sites. The accumulated MCM induces new replication initiation as shown in Fig.5 and Fig. S6. We added the sentence "The trapped MCM would accumulate and may induce DNA replication initiation in the arrival direction of transcription upstream of the barrier" in the revised main text to clarify this.

With regards to the experiment on *GALNT10*, we performed a new repeat and now showed the (mean \pm standard error) curve from three repeats in Fig. S6h.

- Figure 7a and b) is used to suggest that transcription-replication collision in gene bodies in DRB-treated cells leads to DNA damage. To make this statement, the authors must 1) confirm the effect by using amanitin, and 2) address if the DNA damage detected depends on replication-transcription collision. The damage is clearly S phase -specific, but it could result from other mechanisms than collision, such as lack of production of histones or other replication factors. RNase H could be used to address dependency on transcription assuming

that R-loops play a role in replication-dependent damage generation upon transcription inhibition. Another approach would be to test if DRB/amanitin produce less damage in late S phase where predominantly non-transcribed regions are replicated.

Thanks for the constructive comments. Since α -amanitin-mediated transcription inhibition is irreversible while DRB is reversible (Bensaude, 2011), therefore, we only used DRB to perform that experiment. Following this very helpful suggestion, we have overexpressed GFP-RNase H1 in the DRB-treated K562 cells. The result showed that the number of γ -H2AX foci was significantly reduced after RNaseH1 overexpression (Fig. 6d and 6e), which supports the DNA damage depends on R-loops.

- Model in Fig 7c. Although intuitive from the existing published models the presented model is not fully supported by the data shown. Re-distribution of ERIZ (and pre-RCs) is shown, but not pushing of pre-RCs by transcription machines. Alternative explanations like dissociation of pre-RCs are not discussed. Sliding of pre-RCs on naked DNA was shown by the Diffley, Remus and Speck labs, but whether it occurs on chromatin less clear.

Thanks for the comments. To support the model, we first showed a redistribution of early replication upon transcription perturbation in three different ways (Fig. 3). Then we detected the redistribution of MCM from non-transcribed regions to transcribed regions while the total amount of chromatin-bound MCM is constant before or after transcription inhibition (Fig. 4). Lastly, we used the dCas9 to trap MCM to induce early DNA replication initiation on gene bodies without ORC binding, which indicates the association of MCM with RNA polymerase II in the transcribed regions (Fig. 5). The data from Fig. 5 strongly support our model and we moved the model to Fig. 5g in the revised manuscript.

We agree that we still cannot fully exclude the possibility that RNA polymerase II dissociates MCM in the transcribed regions while new MCM is loaded into non-transcribed regions to keep the constant chromatin-bound amount. We added this into the Discussion in the revised manuscript.

- Discussion:

The discussion is very short and should more comprehensively discuss the presented experiments in light of the existing literature.

Thanks. We have extended the “Discussion” section in light of the present literature, including the pros and cons of NAIL-seq, the transcription bull-dozing model and its biological significance. We also added several specific points as suggested to cover previous findings.

References

Bensaude, O. (2011). Inhibiting eukaryotic transcription: Which compound to choose? How to evaluate its activity? *Transcription* 2, 103-108. 10.4161/trns.2.3.16172.

Powell, S.K., MacAlpine, H.K., Prinz, J.A., Li, Y., Belsky, J.A., and MacAlpine, D.M. (2015). Dynamic loading and redistribution of the Mcm2-7 helicase complex through the cell cycle. *EMBO J* 34, 531-543. 10.15252/embj.201488307.

Reviewer #2: Liu et al. GBIO-D-20-02050

"Transcription shapes DNA replication initiation to preserve genome integrity"

In this manuscript, Liu et al. explore the link between transcription and replication origin activity in human cells -- specifically, the long-standing observation that origin activity in early S phase is reduced or absent in actively transcribed regions. The first part of the manuscript focuses on establishing their genome-wide replication assay, pulse-labeling with nucleoside analogues followed by sequencing of the newly replicated DNA, in two human cell lines. They find that early origin activity is excluded in actively transcribed genes but not in transcriptionally silent genes, a correlation that is supported by epigenetic marks of active vs. silent chromatin. Of particular interest is that H2A.Z very strongly correlates with early initiation sites, again consistent with the recent literature on the role of H2A.Z in recruiting ORC1.

This portion of the manuscript mostly recapitulates the previously documented negative correlation between transcription and early origin firing, suggestive of a mechanism where active transcription prevents origin firing. What is newer is that the authors then go on to perturb transcription *in vivo* to test the specific hypothesis that active transcription pushes assembled MCM complexes out of the transcribed regions so that they end up in non-transcribed regions. Initiation of DNA synthesis, which occurs at the MCM complexes, would therefore be excluded from transcribed regions. They use three approaches to test this hypothesis -- inhibition of polIII-driven transcription with alpha-amanitin, read-through transcription using a nuclease-deficient allele of XRN2, and blocking transcription using dCas9. In all three, they examine origin activity as well as locations of ORC and MCM. Their model predicts that limiting transcription should expand the zones of early origin firing, whereas read-through transcription should further restrict early replication initiation sites.

For the most part, their results are consistent with their model. However, there were several areas where they did not provide sufficient detail to adequately evaluate the data (e.g., the pairs of tandem genes they looked at in their read-through transcription experiment), and there were controls that should be included that either weren't done or weren't shown (e.g., validating the transcription block; see specific comments below). In fact, their manuscript overall needs more detail on logic and expectations -- e.g., stating more clearly what outcomes they expect to see (with regard to the figures they show) if their hypothesis is correct vs. if it is not. If suitably revised and with the additional controls, this work can be a nice addition to existing literature on transcription vs. origin activation.

We appreciate the reviewer's constructive comments, which help us to greatly improve our manuscript. Briefly, we have reorganized the figures and added several pieces of new data to support our conclusions including IAA-mediated specific degradation of RNA polymerase II and the RNaseH1 overexpression experiment. We rewrote many sentences and paragraphs to make the manuscript easy to follow. We also expanded the Discussion to cover more present literature. We hope the reviewer finds the revised manuscript acceptable for publication in *Genome Biology* now.

Specific comments (in no particular order)

1. The authors make much of the EdU/BrdU-seq method, but it is not at all clear what the

benefit of this method is over the more conventional EdU-seq. The best resolution they get is either by doing EdU/HU-seq (as has been done by others in the past) or by subtracting the BrdU signal from the EdU+BrdU signal... in other words, they seem to be adding BrdU signal and then subtracting it out, which just seems to be introducing unnecessary complexity and data processing. Why not just do a short EdU pulse? More explanation/justification is needed.

Thanks for the comments. As indicated, the EdU/HU shows a higher resolution than the E-B signals. However, HU has the potential in inducing DNA damages or early firing of late origins as in the literature or we found in this study. The E-B signals are from cells under G1/S transition without replication stress and should not contain replication signals from late origins. The overlapped analysis help remove late origins from EdU/HU samples by E-B samples and we found the non-ERIZs (EdU/HU peaks not overlapped with E-B peaks) showed a later replication timing (Fig. 1e). We revised the sentence as “HU treatment yields high-resolution replication-associated peaks but may induce extra DNA double-stranded breaks (DSBs) and early utilization of late DNA replication origins [39-42]; while the E-B signals are from cells under G1/S transition without replication stress, therefore E-B signals would help discriminate early firing of late origins and potential DNA damages in the EdU/HU libraries” to make this point clear.

We have tested the impact of incorporation time on the resolution of the EdU-seq with a 5, 15, or 60-min incubation of EdU without HU treatment. Different treatments map the same early replication regions with comparable resolutions, not improved gradually as anticipated (see figure below). Under HU treatment, a short pulse is not sufficient to incorporate enough EdU into the genome for library preparation.

2. On a related note, they show comparisons of their data with repli-seq and Okazaki-seq from the literature. However, they don't show comparisons with EdU/HU-seq (e.g., from the Macheret and Halazonetis paper, ref. 14), which would seem the closest comparison. They should include such comparisons.

Thanks for this suggestion. We compared the EdU/HU-seq data with the data from Tubbs et al., Cell, 2018 in activated primary B cells (Tubbs et al., 2018). About 98.4% of identified EdU/HU peaks are overlapped with their EdU/HU peaks (Fig. S2e). With regards to the human cells, we performed EdU/HU-seq in the K562 and GM12878 cells while Drs. Macheret and Halazonetis performed their analysis in the U2OS cells (Macheret and Halazonetis, 2018). Since transcription shapes DNA replication initiation, different transcription profiles in these cells lead to varied patterns of ERIZs. Only 57% of identified ERIZs in GM12878 cells are overlapped with 66% ERIZs from K562 cells, though both of them are immune cell lines.

3. In their EdU/HU-seq plots (e.g., Fig. 1b), the peaks appear to be split, as though the signal is representing forks that have already diverged (ie., forks at the two ends of the replication bubble). Is that the appropriate interpretation? If so, are they not capturing initiation events early enough? More explanation is needed. Also, at the bottom of p.5, the authors note that multiple SNS-seq peaks fell into a single early replication initiation zone identified in this

manuscript (Fig. S2a). They should mark those locations clearly in the figure, or give the coordinates they are talking about in the text, so that the reader can see which peaks they are describing.

The diverged peaks in one ERIZs represent different replication origins close to each other but are unlikely elongation forks. First, EdU is added at the release from G1 in the EdU-seq-HU samples and the replication signals from origins would not be omitted. Second, we have tested the concentrations of HU for EdU-seq-HU and found that the diverged peaks are completely separated from each other under higher concentrations of HU (black triangles in the below figure), indicating that they are independent replication origins. However, higher concentrations of HU lead to a dramatic decrease of replication signals and may show more undesired effects, so we used 10 mM HU as previously used (Tubbs *et al.*, 2018).

Thanks for these nice suggestions. We highlighted the regions of multiple SNS-seq peaks in one ERIZ in Fig. S2a and emphasized them in the main text and figure legends in the revised manuscript.

4. A minor point, they refer to "peak length" in figures 1c and S2b. Perhaps "peak width" would be a more conventional term.

Thanks for the suggestion. We used "peak width" instead of "peak length" in the revised manuscript.

5. In some figures, graphs that we should be able to compare directly are plotted on different scales, which can be misleading (e.g., Fig. 3b and c, Fig 5f). Plots should be on the same scale. For example, if control and amanitin-treated samples in Fig. 5f are plotted on the same scale, the apparent difference might disappear or not be as pronounced. In fact, looking at the summary plot in Fig. 5g, it does appear that the difference, although significant, is slight.

We apologize for the potential misleading. In the revised manuscript, we have shown the plots on the same scale and the conclusions are the same. With regards to MCM (previous Fig. 5f, 5g, now Fig. 4d, 4e), we used Z-score to show the changes of MCM in the revised manuscript and the changes in the non-transcribed regions are still obvious. There is a 2.2-fold change of non-transcribed over transcribed regions after transcription perturbation (the y axis is in the log scale).

6. In figures 4 and 5 (looking at 410 pairs of co-directionally transcribed genes), it is not clear what the distribution of spacing in the gene pairs is. They should provide a histogram of the inter-gene distances, as this distribution affects the interpretation of the data. For example, in the polII ChIP-seq profiles in Fig 4e, what leads to the very distinct hills-and-

valleys shape downstream of the TTS? In how many of the 410 pairs is the second gene beyond the 150 kb region that is plotted, and in how many is the second gene (transcribed region) within the 150 kb downstream of the TTS that is shown?

We showed the distribution of inter-gene distances in the attached figure under Comment 7 and the 410 pairs of genes lie between 20 and 100 kb. Therefore, the distinct hills-and-valleys in the RNA polymerase II ChIP-seq profiles are indeed caused by the TSS of downstream genes as the reviewer indicated.

7. Since they must already have the data, they should also show, for comparison, the polII and early replication data for convergently transcribed genes (where there should be a shift) and in oppositely-oriented genes (aligned by TSS) where according to their model there should not be a shift. Given the small shifts seen in their data, having these additional controls would be helpful. In fact, in Fig. S5g it appears that read-through transcription shifts the bulk of the EdU replication signal towards the gene and not away from the gene as would be expected of their model.

Thanks for the constructive comments. We have analyzed the distributions of RNA polymerase II and early replication initiation signals at convergent or divergent gene pairs. We found a synergistic shift between RNA polymerase II and early replication initiation at both convergent and divergent gene pairs. The shifts of RNA polymerase II and early replication initiation at the divergent gene pairs are probably due to that over 77% of transcription start sites (TSS) supports divergent transcription (Core et al., 2008) and the disturbance of XRN2 also allows transcription read-through of TSS towards the intergenic regions (Brannan et al., 2012).

In the original Fig. S5g, the replication initiation signals are reduced in the downstream intergenic regions of *RAPGEFL1* after transcription read-through. However, there are higher replication initiation signals on the gene body of *WIPF2* due to transcription read-through, despite that the change is mild. As suggested by reviewer 1 and the editor, we have removed the transcription readthrough experiment data from the revised manuscript.

8. Fig. 4a and 4c are not very helpful and can be eliminated.

Thanks. We removed Fig.4 and condensed Fig. 2&3 in the revised manuscript to keep the revised manuscript more focused.

9. Fig. 5b -- why is the EdU replication signal more tightly resolved in the amanitin-treated samples? Is this a consequence of slower forks, or a change in S phase progression? Or is it because the control (untreated) plot is not to the same color scale as the experimental sample plots? More explanation and setup (clear statement of expectations) is needed. Also, for Fig. 5f and g, they should say more clearly what "fold change" means.

Thanks for the comments. The more tightly resolved EdU signal in the original Fig 5b is due to that the control samples were sequenced deeper than the treated samples (12M vs 4 or 5M). We now used similar total reads (4M vs 4 or 5M) and plotted them on the same color scale in Fig. 3a in the revised manuscript and their resolutions look similar to each other. We also performed the DNA fiber assay to analyze the replication speed with or without α -amanitin treatment and found that α -amanitin treatment did not significantly slow the speed of replication forks (see figure below). As showed in Fig. S4b, the S phase progression only showed a slight decrease in the presence of α -amanitin (from 48% to 44% of the early S phase cells).

In Fig. 5f and g, "fold change" means the RPKM ratios of MCM5 ChIP-seq over the input sample. A similar strategy has been employed by *ENCODE* project and a report (Powell *et al.*, 2015) to deal with the MCM ChIP-seq data.

10. The authors should consider showing a plot of the difference between the ORC signal and MCM signal. If ORC is unperturbed but MCM does shift, we should see a greater separation between the MCM and ORC signal locations in the perturbed vs. control (unperturbed).

Thanks for the comments. We performed the analysis as suggested. As also indicated by Reviewer 1, the excess amount of MCM on the chromatin (active and inactive) may blur the signal change. However, as shown below, in the untreated samples, MCM distributes widely within no specific accumulation at ± 50 kb regions within the ORC binding sites; while in the treated samples, we detected a sudden decrease of MCM right at the ORC binding sites and the perfect docking pattern indicates the enrichment of MCM at the ORC-adjacent regions. The MCM is showed via Z-score which means the pattern but not the value can be compared between two samples.

11. For the transcription blockage using dCas9, just showing the overall level of transcription does not adequately validate that the block is working as expected. They need to show RT-qPCR signal (or polII occupancy) to the left vs. the right of the block to show that there is no change upstream of the block vs. reduced transcription downstream of the block. "Relative transcription level" (Fig. 6b) is cryptic, it is not clear relative to what. It is unclear how the quantification for Fig. 6g was done, more detail is needed. Also, in Fig. 6d and f, the symbol for gRNA-CMIP is missing in the legends.

We appreciate the reviewer's comments. We examined the transcription levels upstream of the two dCas9-binding genes and added the new data in the revised manuscript (Fig. S6a, S6b, and S6e). The transcription levels upstream are constant with or without the dCas9 blockade.

We apologized for the confusion. The "relative transcription level" in the original Fig. 6b is relative to the transcription level of *CMIP* without dCas9 blockade. We have added that in the revised figure legends.

The quantification data from Fig. 6g is obtained as follows: the input DNA or ChIP-ed DNA against MCM is subjected to qPCR, the Cq value was used to quantify the amount of DNA, then the percentage of ChIP-ed DNA over input DNA was shown. The quantification data were not from the bands in the gel. The gel only showed that the primer sets obtained the expected DNA bands. To avoid potential misleading, we have removed the gel image from the revised manuscript.

We also have added the description of gRNA-CMIP in the revised figure legends of Fig. 5b and 5d.

12. The authors should consider moving the microscopy images in Fig. S9 to the main figure (Fig. 7).

Thanks for the suggestions. We have moved the microscopy images to the main figure.

13. In the Introduction (p.4, lines 86-88) the authors state that "...suppression of RNA polymerase I, the primary ribosomal DNA transcription polymerase, also causes MCM relocation at the rDNA locus in budding yeast [38]." That statement is not correct, what that paper showed was that suppression of silencing of polII-driven transcription in the rDNA spacer led to MCM relocation.

We have corrected the statement as "RNA polymerase II is also able to cause MCM relocation at the ribosomal DNA loci in budding yeast" in the revised manuscript.

14. On p.6, line 145: "precious" should be "previous".

We have corrected the error in the revised manuscript.

15. On the same page, line 156, and Fig. 2: it is unclear what the authors mean by "intra-compartment activity". Are they talking about the level of transcription within the compartment? Or the frequency of contacts within the compartment in 3D space? "The

levels of ERIZ-associated early replication ranked from low to high in the same order as the compartment activity and transcription level" (lines 157-159) implies that "compartment activity" is a separate phenomenon than transcription. But, "These data indicate that transcription in the active A compartments is coincident with early DNA replication initiation" (lines 161-162) suggests that "intra-compartment activity" refers to transcription activity. Again, more explanation is needed.

We apologize for the confusion. The "intra-compartment activity" is determined by the value of the eigenvector from in-situ Hi-C data (Rao et al., 2014). The eigenvector is defined by principal component analysis on the Hi-C interaction matrix, while the positive value of the eigenvector correlates with the active chromatin compartment (Lieberman-Aiden et al., 2009). The absolute value of the eigenvector correlates with the activeness of the compartment (Miura et al., 2019). We categorized all the A compartments into four groups according to the mean value of eigen vector in each compartment. The transcription activity and early replication initiation level show the same trends as the eigen vector.

Reference

Bensaude, O. (2011). Inhibiting eukaryotic transcription: Which compound to choose? How to evaluate its activity? *Transcription* 2, 103-108. 10.4161/trns.2.3.16172.

Brannan, K., Kim, H., Erickson, B., Glover-Cutter, K., Kim, S., Fong, N., Kiemele, L., Hansen, K., Davis, R., Lykke-Andersen, J., and Bentley, D.L. (2012). mRNA decapping factors and the exonuclease Xrn2 function in widespread premature termination of RNA polymerase II transcription. *Mol Cell* 46, 311-324. 10.1016/j.molcel.2012.03.006.

Core, L.J., Waterfall, J.J., and Lis, J.T. (2008). Nascent RNA sequencing reveals widespread pausing and divergent initiation at human promoters. *Science* 322, 1845-1848. 10.1126/science.1162228.

Lieberman-Aiden, E., van Berkum, N.L., Williams, L., Imakaev, M., Ragozy, T., Telling, A., Amit, I., Lajoie, B.R., Sabo, P.J., Dorschner, M.O., et al. (2009). Comprehensive mapping of long-range interactions reveals folding principles of the human genome. *Science* 326, 289-293. 10.1126/science.1181369.

Macheret, M., and Halazonetis, T.D. (2018). Intragenic origins due to short G1 phases underlie oncogene-induced DNA replication stress. *Nature* 555, 112-116. 10.1038/nature25507.

Miura, H., Takahashi, S., Poonperm, R., Tanigawa, A., Takebayashi, S.I., and Hiratani, I. (2019). Single-cell DNA replication profiling identifies spatiotemporal developmental dynamics of chromosome organization. *Nat Genet* 51, 1356-1368. 10.1038/s41588-019-0474-z.

Powell, S.K., MacAlpine, H.K., Prinz, J.A., Li, Y., Belsky, J.A., and MacAlpine, D.M. (2015). Dynamic loading and redistribution of the Mcm2-7 helicase complex through the cell cycle. *EMBO J* 34, 531-543. 10.15252/embj.201488307.

Rao, S.S., Huntley, M.H., Durand, N.C., Stamenova, E.K., Bochkov, I.D., Robinson, J.T., Sanborn, A.L., Machol, I., Omer, A.D., Lander, E.S., and Aiden, E.L. (2014). A 3D map of the human genome at kilobase resolution reveals principles of chromatin looping. *Cell* 159, 1665-1680. 10.1016/j.cell.2014.11.021.

Tubbs, A., Sridharan, S., van Wietmarschen, N., Maman, Y., Callen, E., Stanlie, A., Wu, W., Wu, X., Day, A., Wong, N., et al. (2018). Dual Roles of Poly(dA:dT) Tracts in Replication Initiation and Fork Collapse. *Cell* 174, 1127-1142 e1119. 10.1016/j.cell.2018.07.011.

Second round of review

Reviewer 1

The revised and more focused manuscript is much clearer than the originally submitted version. The authors focus now on A) the consequences on early replication by globally inhibiting transcription, and B) on the consequences of putting a transcription roadblock into the body of a specific gene. For point A) they added the approach of inhibiting transcription by auxin-mediated degradation of RNA polymerase to confirm results by chemical inhibitors.

I think the collection of data presented is now generally suitable for publication. I described the relevance of the findings earlier. However, a few important points have still to be addressed. These points include clarity of presentation and data display (points 1-3), and interpretation of observed effects including the models concluded (points (3), 4 and 5).

Major points to address:

1) The language used is unclear. Because this compromises readability and clarity I advise the consultation an English native speaker to smoothen the language before publication.

2) Characterisation of NAIL.

I expect a clear statement if any (and how much) replication is found in typical late replicating genome regions.

3) Figure 3

- Because not initiation is measured directly, the title of the figure should be changed into "Transcription relocated early replication in K562 and mESC"

- I like the zoom-in view for pol II depletion presented in 3g. It gives a detailed view that enriches and complements the genome-wide data with higher statistical power shown in the other panels. Similar zoom-in views should be presented for alpha amanitin and DRB to be consistent with the presentation of the data. Such consistence aids com[arability of the results for the different approaches.

- As suggested in my original review I think that a statement about the correlation between the strength of inhibition of transcription and replication re-location should be presented to provide a better impression of the results presented.

- There is a striking difference in localisation of early replication between amanitin (distinct localization at TSS) on the one side, and DRB/PolII depletion (increase in gene bodies) on the other side. These observations are not contradictory, because they are both consistent with re-localisation away from non-transcribed regions between genes. However, the differences in the observations have to be rationalised and explained. I suppose the reason lies in mechanistic differences of the different treatments.

4) Fig 5:

- Panels a and g are partly contradictory and that contradiction must be eliminated from the figure. a and g point to a major issue raised in my review of the original manuscript: a indicates that pre-RCs accumulate at the block (or slightly upstream). a ignores that pre-RCs downstream of the block are not eliminated from the gene. a is used to indicate that early replication could be enhanced slightly upstream of the block, consistent with the results of panel b. In contrast, g ignores the potential accumulation of pre-RCs at the block but shows that pre-RCs are not cleared downstream of the block, suggesting that early replication is now expected downstream of the block, which cannot be seen in the modified gene in panel b.

Please clearly state where in the gene early replication can be expected upon blocking transcription, namely potentially upstream very close to the block and downstream of the block, but not further upstream closer to the TSS (where transcription has cleared pre-RCs). Perhaps, a stronger increase is expected close to the block, where potentially all pre-RCs from upstream have accumulated, whereas lack of clearance but no accumulation of pre-RCs is expected downstream of the block. Please present an explanation for why increased early replication upstream of the block but not downstream of the block is found in panel b. Perhaps, in the particular gene there are no/few pre-RCs downstream the block.

5) Model of transcription shaping early replication

I think alternative models of transcription shaping replication by pushing pre-RCs are not appropriately discussed. I agree that the pushing model may be the most likely model in light of the literature and the data presented, but it is not proven, because: 1) the pushing process itself has not been observed in this paper (which would be very difficult), 2) The evidence that early replication upstream of the Cas9 block is by firing of relocated pre-RCs is scarce. Such an alternative model is that transcription dissociates pre-RCs from genes.

Please present arguments for and against the models in a structured way, including a) can pre-RCs slide on DNA and under which conditions?, can they slide on chromatin?, what has been shown regarding pre-RC pushing by transcription in vitro and in budding yeast cells?

In my view, the observation of replication accumulating at the Cas9 block hints at the pushing model because a dissociation model does not predict accumulation of pre-RCs at the block and therefore predicts initiation downstream the block due to lack of pre-RC dissociation because transcription machines have not passed through.

In line with this, please change the statement in line 109: "... we found that RNA

105 polymerase II can actively push loaded MCM but not ORC outside of transcribed genes...". It is too strong and ignores alternative explanations.

Further points:

Fig. 4:

- It seems that panel f indicates that, despite a generally low level of ERIZ in gene bodies, individual genes seem to have a lot of ERIZ (red lines in the upper part control-treated sample). Could it be interesting to look at these genes? Are they low-transcribed? Do they contain hot spots for genetic changes?

line 128: I do not understand the sentence "The EdU or BrdU..." in this context.

line 139: change to "...and utilization of dormant DNA replication origins"

line 211: The sentence "Moreover, the early replication signals showed a significant enrichment at the transcription start sites (TSSs) following alpha-amanitin treatment (Figs. 3a and 4f)" could be misleading, because the expectation is that early replication is in gene bodies not at TSSs. Please elaborate to clarify.

Reviewer 2

The authors have largely addressed my concerns and the manuscript is much improved. The degon experiment is a nice addition that supports their conclusions. There are a couple of points that the authors have somewhat addressed but still leave me with lingering questions, but these are relatively minor.

1. On my previous question regarding split peaks: the authors have responded that because they are adding EdU and HU at the time of release from G1, closely-spaced peaks must represent separate initiation sites and not forks at the ends of a bubble. However, it seems to me (as a non-expert in this technique) that if there is a lag between the time when cells encounter EdU and when endogenous dTTP pools are depleted so that the EdU is efficiently incorporated in DNA, the earliest origins could fire without EdU incorporation at the precise initiation site. (One way around this problem would be to add EdU some time before release from G1, allowing time for intracellular pools to equilibrate.) If this scenario is not a worry it would help if the authors would explain either in Methods or when they first describe their technique why adding EdU at the time of release (rather than prior to release) will adequately capture the earliest initiation events.

2. Regarding transcription blockage using dCas9: It is reassuring that transcript levels upstream of the blockage (Primers P1+P2) are unchanged while overall transcript levels are reduced. So, if I understand the experiment right, they are providing indirect evidence that the effect of the block is seen only downstream. In my mind, the cleanest evidence would be to show that ratio of P1+P2 vs. P5+P6 with and without the block.

3. Lines 311-312: the sentence is incomplete. Perhaps the authors mean to say, "It is conceivable that early DNA initiation from transcribed regions..."?

4. Likewise, lines 332-337: The sentence begins, "While the overexpression of RNase H1..." but that thought does not seem to be completed. Maybe split that long sentence into two, so that the logic is easier to understand?

5. Line 328: a typo, "asynchronzided" -- should be "asynchronized"

6. Line 363: Given the caveat later in the paragraph that RNA PolIII may cause dissociation of MCM complexes from chromatin, it may be more accurate to say that the authors have "concluded" that RNAP II acts as a bulldozer, rather than saying that they "found" that it acts as a bulldozer. The distinction is subtle, but "found" implies to me that there is direct evidence, whereas "concluded" more accurately reflects the fact that the bulldozer idea is one (likely) interpretation of the data.

Authors Response

Point-by-point responses to the reviewers' comments:

Reviewer #1: The revised and more focused manuscript is much clearer than the originally submitted version. The authors focus now on A) the consequences on early replication by globally inhibiting transcription, and B) on the consequences of putting a transcription roadblock into the body of a specific gene. For point A) they added the approach of inhibiting transcription by auxin-mediated degradation of RNA polymerase to confirm results by chemical inhibitors.

I think the collection of data presented is now generally suitable for publication. I described the relevance of the findings earlier. However, a few important points have still to be addressed. These points include clarity of presentation and data display (points 1-3), and interpretation of observed effects including the models concluded (points (3), 4 and 5).

Thanks for your comments and suggestions. We believe that they helped us greatly with regard to improving our manuscript.

Major points to address:

1) The language used is unclear. Because this compromises readability and clarity I advise the consultation an English native speaker to smoothen the language before publication.

Thank you for the comment. The revised manuscript has been polished by a native speaker from a commercial service. We hope that the revised manuscript is easier for readers to follow.

2) Characterisation of NAIL.

I expect a clear statement if any (and how much) replication is found in typical late replicating genome regions.

We found that 0.04% (1 of 2,265) and 1% (30 of 2,874) of ERIZs overlapped with typical late replication regions in K562 and GM12878 cells, respectively. We have described these results in the revised manuscript.

3) Figure 3

- Because not initiation is measured directly, the title of the figure should be changed into "Transcription relocated early replication in K562 and mESC"

Thank you for the comment. We have removed the word "initiation" from the title of Figure 3.

- I like the zoom-in view for pol II depletion presented in 3g. It gives a detailed view that enriches and complements the genome-wide data with higher statistical power shown in the other panels. Similar zoom-in views should be presented for alpha amanitin and DRB to be consistent with the presentation of the data. Such consistence aids comparability of the results for the different approaches.

Thank you for the comment. We have included zoomed-in views for α -amanitin and DRB treatment in the revised manuscript (Additional file 1: Fig. S4d and S4g).

- As suggested in my original review I think that a statement about the correlation between the strength of inhibition of transcription and replication re-location should be presented to provide a better impression of the results presented.

Thank you for the nice comment. We have included the statement that "Moreover, stronger inhibition of transcription resulted in more dramatic early replication redistribution on the genome (Fig. 3b and exemplified in Additional file 1: Fig. S4d)" in the revised manuscript. As you may find in Fig. S4d,

the higher tested concentration of α -amanitin led to a more obvious shift of ERIZs into the transcribed *TRPC4AP* gene.

- There is a striking difference in localisation of early replication between amanitin (distinct localization at TSS) on the one side, and DRB/PolIII depletion (increase in gene bodies) on the other side. These observations are not contradictory, because they are both consistent with re-localisation away from non-transcribed regions between genes. However, the differences in the observations have to be rationalised and explained. I suppose the reason lies in mechanistic differences of the different treatments.

Thank you for the comment. α -Amanitin is an inhibitor of Pol II transcription initiation that suppresses Pol II binding at promoters (or TSS) and gene bodies. DRB is an inhibitor of transcription elongation that decreases Pol II binding at gene bodies, but increases Pol II binding at promoters (or TSS). The degron system partially blocks, but does not completely block, Pol II binding at both gene bodies and promoters (or TSS). Therefore, α -amanitin may have a stronger impact on the TSS in comparison with the other two treatments, but it may be that some unexplored mechanism of α -amanitin is also involved in this effect. As shown in the newly added Fig. S4d, we detected robust signals at ORC-resident TSSs in the displayed region. Moreover, enrichment at TSSs following α -amanitin treatment was also observed in samples from mouse primary B cells (Additional file 1: Fig. S4e). We wish that we could provide a better answer, but we have to admit that we have no definite answer for this phenomena. Thanks again for the insightful comment.

4) Fig 5:

- Panels a and g are partly contradictory and that contradiction must be eliminated from the figure. a and g point to a major issue raised in my review of the original manuscript: a indicates that pre-RCs accumulate at the block (or slightly upstream). a ignores that pre-RCs downstream of the block are not eliminated from the gene. a is used to indicate that early replication could be enhanced slightly upstream of the block, consistent with the results of panel b. In contrast, g ignores the potential accumulation of pre-RCs at the block but shows that pre-RCs are not cleared downstream of the block, suggesting that early replication is now expected downstream of the block, which cannot be seen in the modified gene in panel b.

Thanks for the comment. We have modified the schematic in panel a to include ORC at downstream initiation sites to emphasize that ORC is crucial for DNA replication initiation within the region downstream from the transcription blockade site.

Please clearly state where in the gene early replication can be expected upon blocking transcription, namely potentially upstream very close to the block and downstream of the block, but not further upstream closer to the TSS (where transcription has cleared pre-RCs). Perhaps, a stronger increase is expected close to the block, where potentially all pre-RCs from upstream have accumulated, whereas lack of clearance but no accumulation of pre-RCs is expected downstream of the block. Please present an explanation for why increased early replication upstream of the block but not downstream of the

block is found in panel b. Perhaps, in the particular gene there are no/few pre-RCs downstream the block.

Thanks for the comment. Upon transcription blockade, MCM complexes accumulate at the block (or slightly upstream) without bound ORC, which results in early replication at the block (or slightly upstream). Regarding the region downstream of the block, ORC is essential for MCM loading to achieve substantial DNA replication initiation. As the reviewer noticed, ORC is largely absent in the region downstream of the block site within *CMIP*. We have included these statements in the Results and Discussion sections of the revised manuscript.

5) Model of transcription shaping early replication

I think alternative models of transcription shaping replication by pushing pre-RCs are not appropriately discussed. I agree that the pushing model may be the most likely model in light of the literature and the data presented, but it is not proven, because: 1) the pushing process itself has not been observed in this paper (which would be very difficult), 2) The evidence that early replication upstream of the Cas9 block is by firing of relocated pre-RCs is scarce. Such an alternative model is that transcription dissociates pre-RCs from genes.

Please present arguments for and against the models in a structured way, including a) can pre-RCs slide on DNA and under which conditions?, can they slide on chromatin?, what has been shown regarding pre-RC pushing by transcription in vitro and in budding yeast cells?

In my view, the observation of replication accumulating at the Cas9 block hints at the pushing model because a dissociation model does not predict accumulation of pre-RCs at the block and therefore predicts initiation downstream the block due to lack of pre-RC dissociation because transcription machines have not passed through.

Thank you for the comment. We agree with the reviewer that we still cannot fully exclude the possibility that RNA polymerase II induces MCM disassociation. In light of this, we have presented arguments for and against the models in a structured way in the Discussion section and have tuned down our statements as suggested.

In line with this, please change the statement in line 109: "... we found that RNA

105 polymerase II can actively push loaded MCM but not ORC outside of transcribed genes...". It is too strong and ignores alternative explanations.

Thank you for the comment. We have changed the sentence into "Furthermore, inhibition of transcription leads to MCM redistribution and early replication, but not ORC re-localization, in transcribed regions".

Further points:

Fig. 4:

- It seems that panel f indicates that, despite a generally low level of ERIZ in gene bodies, individual genes seem to have a lot of ERIZ (red lines in the upper part control-treated sample). Could it be interesting to look at these genes? Are they low-transcribed? Do they contain hot spots for genetic changes?

We have looked at these genes. They are generally short in length (the genes are aligned from short to long with the shortest on the top), and the early replicated neighbors might penetrate into the borders of their gene bodies.

line 128: I do not understand the sentence "The EdU or BrdU..." in this context.

We have changed the sentence to "Moreover, the "E-B" signal obtained by subtracting the BrdU signal from the EdU signal was further narrowed to the middle of the early replication domains (Fig. 1b; Additional file 1: Fig. S2a)".

line 139: change to "...and utilization of dormant DNA replication origins"

We have changed the word as suggested.

line 211: The sentence "Moreover, the early replication signals showed a significant enrichment at the transcription start sites (TSSs) following alpha-amanitin treatment (Figs. 3a and 4f)" could be misleading, because the expectation is that early replication is in gene bodies not at TSSs. Please elaborate to clarify.

We rewrote the sentence as "Early replication signals also showed increased enrichment at transcription start sites (TSSs) following α -amanitin treatment (Figs. 3a and 4f; Additional file 1: Fig. S4d), possibly because TSS regions where ORC is located support DNA replication initiation, as previously reported [10, 12, 23, 48]".

The authors have largely addressed my concerns and the manuscript is much improved. The degranulation experiment is a nice addition that supports their conclusions. There are a couple of points that the authors have somewhat addressed but still leave me with lingering questions, but these are relatively minor.

1. On my previous question regarding split peaks: the authors have responded that because they are adding EdU and HU at the time of release from G1, closely-spaced peaks must represent separate initiation sites and not forks at the ends of a bubble. However, it seems to me (as a non-expert in this technique) that if there is a lag between the time when cells encounter EdU and when endogenous dTTP pools are depleted so that the EdU is efficiently incorporated in DNA, the earliest origins could fire without EdU incorporation at the precise initiation site. (One way around this problem would be to add EdU some time before release from G1, allowing time for intracellular pools to equilibrate.) If this scenario is not a worry it would help if the authors would explain either in Methods or when they first describe their technique why adding EdU at the time of release (rather than prior to release) will adequately capture the earliest initiation events.

Thank you for the insightful comments. We used 10 μ M EdU to label nascent DNA, which is comparable with the concentration of endogenous dTTP ($17.6 \pm 7.23 \mu$ M) in K562 cells [1]. Moreover, EdU can be incorporated into nascent DNA in a very efficient way, and 2 minutes are sufficient for EdU to label nascent Okazaki fragments in OK-seq [2]. The G1-released K562 cells required an additional 2.5 hours to reach the G1/S transition. Therefore, initiation sites are unlikely to escape EdU labelling in the EdU/HU assay. A similar strategy has also been employed in EdU-seq-HU in U2OS cells, in which EdU was added 2 hours before entering the S phase [3]. We have included these points in the Methods section.

2. Regarding transcription blockage using dCas9: It is reassuring that transcript levels upstream of the blockage (Primers P1+P2) are unchanged while overall transcript levels are reduced. So, if I understand the experiment right, they are providing indirect evidence that the effect of the block is seen only downstream. In my mind, the cleanest evidence would be to show that ratio of P1+P2 vs. P5+P6 with and without the block.

Thank you for the suggestion. We performed the analysis as suggested and depicted the new results in Fig. S6c. As shown in Fig. S6c, the ratio of P1+P2 over P5+P6 was significantly increased in dCas9-treated cells in comparison with that of cells without dCas9-blockade.

3. Lines 311-312: the sentence is incomplete. Perhaps the authors mean to say, "It is conceivable that early DNA initiation from transcribed regions...?"

Thank you for the comment. We have re-written the sentence as suggested.

4. Likewise, lines 332-337: The sentence begins, "While the overexpression of RNase H1..." but that thought does not seem to be completed. Maybe split that long sentence into two, so that the logic is easier to understand?

Thank you for the comment. We have re-written the sentence in the revised manuscript.

5. Line 328: a typo, "asynchronizded" -- should be "asynchronized"

We have corrected the error in the revised manuscript.

6. Line 363: Given the caveat later in the paragraph that RNA PolIII may cause dissociation of MCM complexes from chromatin, it may be more accurate to say that the authors have "concluded" that RNAP II acts as a bulldozer, rather than saying that they "found" that it acts as a bulldozer. The distinction is subtle, but "found" implies to me that there is direct evidence, whereas "concluded" more accurately reflects the fact that the bulldozer idea is one (likely) interpretation of the data.

Thank you for the comment. We have replaced "found" with "concluded" in the revised manuscript.

References:

1. Chen P, Liu Z, Liu S, Xie Z, Aimiwu J, Pang J, Klisovic R, Blum W, Grever MR, Marcucci G, Chan KK: **A LC-MS/MS method for the analysis of intracellular nucleoside triphosphate levels.** *Pharm Res* 2009, **26**:1504-1515.
2. Petryk N, Kahli M, d'Aubenton-Carafa Y, Jaszczyszyn Y, Shen Y, Silvain M, Thermes C, Chen CL, Hyrien O: **Replication landscape of the human genome.** *Nat Commun* 2016, **7**:10208.
3. Macheret M, Halazonetis TD: **Intragenic origins due to short G1 phases underlie oncogene-induced DNA replication stress.** *Nature* 2018, **555**:112-116.

Third round of review

Reviewer 1

Very good revisions.

The manuscript now describes the research clearly. The experiments are conclusive. The conclusions are sufficiently supported by experimental evidence and convincing.